# LATENT GRAPH INFERENCE
# USING PRODUCT MANIFOLDS

**Haitz Sáez de Ocáriz Borde**
University of Oxford

**Anees Kazi**
Harvard University

**Federico Barbero**
University of Oxford

**Pietro Liò**
University of Cambridge

## ABSTRACT

Graph Neural Networks usually rely on the assumption that the graph topology is available to the network as well as optimal for the downstream task. Latent graph inference allows models to dynamically learn the intrinsic graph structure of problems where the connectivity patterns of data may not be directly accessible. In this work, we generalize the discrete Differentiable Graph Module (dDGM) for latent graph learning. The original dDGM architecture used the Euclidean plane to encode latent features based on which the latent graphs were generated. By incorporating Riemannian geometry into the model and generating more complex embedding spaces, we can improve the performance of the latent graph inference system. In particular, we propose a computationally tractable approach to produce product manifolds of constant curvature model spaces that can encode latent features of varying structure. The latent representations mapped onto the inferred product manifold are used to compute richer similarity measures that are leveraged by the latent graph learning model to obtain optimized latent graphs. Moreover, the curvature of the product manifold is learned during training alongside the rest of the network parameters and based on the downstream task, rather than it being a static embedding space. Our novel approach is tested on a wide range of datasets, and outperforms the original dDGM model.

## 1 INTRODUCTION

Graph Neural Networks (GNNs) have achieved state-of-the-art performance in a number of applications, from travel-time prediction (Derrow-Pinion et al. (2021)) to antibiotic discovery (Stokes et al. (2020)). They leverage the connectivity structure of graph data, which improves their performance in many applications as compared to traditional neural networks (Bronstein et al. (2017)). Most current GNN architectures assume that the topology of the graph is given and fixed during training. Hence, they update the input node features, and sometimes edge features, but preserve the input graph topology. A substantial amount of research has focused on improving diffusion using different types of GNN layers. However, discovering an optimal graph topology that can help diffusion has only recently gained attention (Topping et al. (2021); Cosmo et al. (2020); Kazi et al. (2022)).

In many real-world applications, data can have some underlying but unknown graph structure, which we call a *latent graph*. That is, we may only be able to access a pointcloud of data. Nevertheless, this does not necessarily mean the data is not intrinsically related, and that its connectivity cannot be leveraged to make more accurate predictions. The vast majority of Geometric Deep Learning research so far has relied on human annotators or simplistic pre-processing algorithms to generate the graph structure to be passed to GNNs. Furthermore, in practice, even in settings where the correct graph is provided, it may often be suboptimal for the task at hand, and the GNN may benefit from rewiring (Topping et al. (2021)). In this work, we drop the assumption that the graph adjacency matrix is given and study how to learn the latent graph in a fully-differentiable manner, using

product manifolds, alongside the GNN diffusion layers. More elaborately, we incorporate Riemannian geometry to the discrete Differentiable Graph Module (dDGM) proposed by Kazi et al. (2022). We show that it is possible and beneficial to encode latent features into more complex embedding spaces beyond the Euclidean plane used in the original work. In particular, we leverage the convenient mathematical properties of product manifolds to learn the curvature of the embedding space in a fully-differentiable manner.

**Contributions**  1) We explain how to use model spaces of constant curvature for the embedding space. To do so, we outline a principled procedure to map Euclidean GNN output features to constant curvature model space manifolds with non-zero curvature: we use the hypersphere for spherical space and the hyperboloid for hyperbolic space. We also outline how to calculate distances between points in these spaces, which are then used by the dDGM sparse graph generating procedure to infer the edges of the latent graph. Unlike the original dDGM model which explored using the Poincaré ball with fixed curvature for modeling hyperbolic space, in this work we use hyperboloids of arbitrary negative curvature. 2) We show how to construct more complex embedding spaces that can encode latent data of varying structure using product manifolds of model spaces. The curvature of each model space composing the product manifold is learned in a fully-differentiable manner alongside the rest of the model parameters, and based on the downstream task performance. 3) We test our approach on 15 datasets which includes standard homophilic graph datasets, heterophilic graphs, large-scale graphs, molecular datasets, and datasets for other real-world applications such as brain imaging and aerospace engineering. 4) It has been shown that traditional GNN models, such as Graph Convolutional Networks (GCNs) (Kipf & Welling (2017)) and Graph Attention Networks (GATs) (Veličković et al. (2018)) struggle to achieve good performance in heterophilic datasets (Zhu et al. (2020)), since in fact homophily is used as an inductive bias by these models. Amongst other models, Sheaf Neural Networks (SNNs) (Hansen & Gebhart (2020); Bodnar et al. (2022); Barbero et al. (2022b;a)) have been proposed to tackle this issue. We show that latent graph inference enables traditional GNN models to give good performance on heterophilic datasets without having to resort to sophisticated diffusion layers or model architectures such as SNNs. 5) To make this work accessible to the wider machine learning community, we have created a new PyTorch Geometric layer.

## 2 BACKGROUND

In this section we discuss relevant background for this work. We first provide a literature review regarding recent advances in latent graph inference using GNNs as well as related work on manifold learning and graph embedding. Next, we give an overview of the original Differentiable Graph Module (DGM) formulation, but we recommend referring to Kazi et al. (2022) for further details.

### 2.1 RELATED WORK

Latent graph and topology inference is a standing problem in Geometric Deep Learning. In contrast to algorithms that work on sets and that apply a shared pointwise function such as PointNet (Qi et al. (2017)), in latent graph inference we want to learn to optimally share information between nodes in the pointcloud. Some contributions in the literature have focused on applying pre-processing steps to enhance diffusion based on an initial input graph (Topping et al. (2021); Gasteiger et al. (2019); Alon & Yahav (2021); Wu et al. (2019)). Note, however, that this area of research focuses on improving an already existing graph which may be suboptimal for the downstream task. This paper is more directly related to work that addresses how to learn the graph topology dynamically, instead of assuming a fixed graph at the start of training. When the underlying connectivity structure is unknown, architectures such as transformers (Vaswani et al. (2017)) and attentional multi-agent predictive models (Hoshen (2017)), simply assume the graph to be fully-connected, but this can become hard to scale to large graphs. Generating sparse graphs can result in more computationally tractable solutions (Fetaya et al. (2018)) and avoid over-smoothing (Chen et al. (2020a)). For this a series of models have been proposed, starting from Dynamic Graph Convolutional Neural Networks (DGCNNs) (Wang et al. (2019)), to other solutions that decouple graph inference and information diffusion, such as the Differentiable Graph Modules (DGMs) in Cosmo et al. (2020) and Kazi et al. (2022). Note that latent graph inference may also be referred to as graph structure learning in the literature. A survey of similar methods can be found in Zhu et al. (2021), and some additional

classical methods include LDS-GNN (Franceschi et al., 2019), IDGL (Chen et al., 2020b), and Pro-GNN (Jin et al., 2020).

In this work, we extend the dDGM module proposed by Kazi et al. (2022) for learning latent graphs using product manifolds. Product spaces have primarily been studied in the manifold learning and graph embedding literature (Cayton (2005); Fefferman et al. (2013); Bengio et al. (2012)). Recent work has started exploring encoding the geometry of data into rich ambient manifolds. In particular, hyperbolic geometry has proven successful in a number of tasks (Liu et al. (2019); Chamberlain et al. (2017); Sala et al. (2018)). Different manifold classes have been employed to enhance modeling flexibility, such as products of constant curvature spaces (Gu et al. (2019)), matrix manifolds (Cruceru et al. (2021)), and heterogeneous manifolds (Di Giovanni et al. (2022)). We will leverage these ideas and use product manifolds to generate the embedding space for constructing our latent graphs.

## 2.2 AN OVERVIEW OF THE DISCRETE DIFFERENTIABLE GRAPH MODULE

Kazi et al. (2022) proposed a general technique for learning an optimized latent graph, based on the output features of each layer, onto which to apply the downstream GNN diffusion layers. Here, we specifically focus on the dDGM module (not the cDGM), which is much more computationally efficient and recommended by the authors. The main idea is to use some measure of similarity between the latent node features to generate latent graphs which are optimal for each layer $l$. We can summarize the architecture as $\hat{\mathbf{X}}^{(l+1)} = f_{\mathbf{\Theta}}^{(l)}(concat(\mathbf{X}^{(l)}, \hat{\mathbf{X}}^{(l)}), \mathbf{A}^{(l)}) \to \mathbf{A}^{(l+1)} \sim \mathbf{P}^{(l)}(\hat{\mathbf{X}}^{(l+1)})) \to \mathbf{X}^{(l+1)} = g_{\phi}(\mathbf{X}^{(l)}, \mathbf{A}^{(l+1)})$. The node features in layer $l$, $\mathbf{X}^{(l)}$, are transformed into $\hat{\mathbf{X}}^{(l+1)}$ through a function $f_{\mathbf{\Theta}^{(l)}}$, which has learnable parameters, and compared using a similarity measure $\varphi(T)$, which is parameterized by a scalar learnable parameter $T$. On the other hand, $g_{\phi}$ is a diffusion function which in practice corresponds to multiple GNN layers stacked together. $g_{\phi}$ diffuses information based on the inferred latent graph connectivity structure summarized in $\mathbf{A}^{(l+1)}$, which is an unweighted sparse matrix. The dDGM module generates a sparse $k$-degree graph using the Gumbel Top-k trick (Kool et al. (2019)), a stochastic relaxation of the kNN rule, to sample edges from the probability matrix $\mathbf{P}^{(l)}(\mathbf{X}^{(l)}; \mathbf{\Theta}^{(l)}, T)$, where each entry corresponds to

$$p_{ij}^{(l)}(\mathbf{\Theta}^{(l)}) = \exp(\varphi(f_{\mathbf{\Theta}}^{(l)}(\mathbf{x}_i^{(l)}), f_{\mathbf{\Theta}}^{(l)}(\mathbf{x}_j^{(l)}); T)) = \exp(\varphi(\hat{\mathbf{x}}_i^{(l+1)}, \hat{\mathbf{x}}_j^{(l+1)}; T)). \tag{1}$$

The main similarity measure used in Kazi et al. (2022) was to compute the distance based on the features of two nodes in the graph embedding space. They assumed that the latent features laid in an Euclidean plane of constant curvature $K_{\mathbb{E}} = 0$, so that $p_{ij}^{(l)} = \exp(-T\mathfrak{d}_{\mathbb{E}}(f_{\mathbf{\Theta}}^{(l)}(\mathbf{x}_i^{(l)}), f_{\mathbf{\Theta}}^{(l)}(\mathbf{x}_j^{(l)}))) = \exp(-T\mathfrak{d}_{\mathbb{E}}(\hat{\mathbf{x}}_i^{(l+1)}, \hat{\mathbf{x}}_j^{(l+1)}))$, where $\mathfrak{d}_{\mathbb{E}}$ denotes distance in Euclidean space. Then, based on $argsort(\log(\mathbf{p}_i^{(l)}) - \log(-\log(\mathbf{q})))$, where $\mathbf{q} \in \mathbb{R}^N$ is uniform i.i.d in the interval $[0, 1]$, we can sample the edges $\mathcal{E}^{(l)}(\mathbf{X}^{(l)}; \mathbf{\Theta}^{(l)}, T, k) = \{(i, j_{i,1}), (i, j_{i,2}), ..., (i, j_{i,k}) : i = 1, ..., N\}$, where $k$ is the number of sampled connections using the Gumbel Top-k trick. This sampling approach follows the categorical distribution $\frac{p_{ij}^{(l)}}{\Sigma_r p_{ir}^{(l)}}$ and $\mathcal{E}(\mathbf{X}^{(l)}; \mathbf{\Theta}^{(l)}, T, k)$ is represented by the unweighted adjacency matrix $\mathbf{A}^{(l)}(\mathbf{X}^{(l)}; \mathbf{\Theta}^{(l)}, T, k)$. Note that including noise in the edge sampling approach will result in the generation of some random edges in the latent graphs which can be understood as a form of regularization. In this work, we generalize Equation 1 to measure similarities based on distances but dropping the assumption used in Kazi et al. (2022), in which they limit themselves to fixed-curvature spaces, specifically to Eucliden space where $K_{\mathbb{E}} = 0$. We will use product manifolds of model spaces of constant curvature to improve the similarity measure $\varphi$ and construct better latent graphs.

## 3 METHOD: PRODUCT MANIFOLDS FOR LATENT GRAPH INFERENCE

In this section, we first introduce model spaces, which are a special type of Riemannian manifolds, and explain how to map Euclidean GNN output features to model spaces with non-zero curvature. In case the reader is unfamiliar with the topic, additional details regarding Riemannian manifolds can be found in Appendix A. Then, we mathematically define product manifolds and how to calculate distances between points in the manifold. Next, we introduce scaling metrics which help us learn

the curvature of each model space composing the product manifold. A discussion on product manifold curvature learning can be found in Appendix B. The intuition behind the method is that we can consider the embedding space represented by the product manifold as a combination of more simple spaces (model spaces of constant curvature), and compute distances between the latent representations mapped onto the product manifold by considering distances in each model space individually and later aggregating them in a principled manner. This allows to generate diverse embedding spaces which at the same time are computationally tractable.

## 3.1 CONSTANT CURVATURE MODEL SPACES

Curvature is effectively a measure of geodesic dispersion. When there is no curvature geodesics stay parallel, with negative curvature they diverge, and with positive curvature they converge. Euclidean space, $\mathbb{E}_{K_{\mathbb{E}}}^{d_{\mathbb{E}}} = \mathbb{R}^{d_{\mathbb{E}}}$, is a flat space with curvature $K_{\mathbb{E}} = 0$. Note that here we use $d_{\mathbb{E}}$ to denote dimensionality. On the other hand, hyperbolic and spherical space, have negative and positive curvature, respectively. We define hyperboloids as $\mathbb{H}_{K_{\mathbb{H}}}^{d_{\mathbb{H}}} = \{\mathbf{x}_p \in \mathbb{R}^{d_{\mathbb{H}}+1} : \langle \mathbf{x}_p, \mathbf{x}_p \rangle_{\mathcal{L}} = 1/K_{\mathbb{H}}\}$, where $K_{\mathbb{H}} < 0$ and $\langle \cdot, \cdot \rangle_{\mathcal{L}}$ is the Lorentz inner product $\langle \mathbf{x}, \mathbf{y} \rangle_{\mathcal{L}} = -x_1 y_1 + \sum_{j=2}^{d_{\mathbb{H}}+1} x_j y_j$, $\forall \mathbf{x}, \mathbf{y} \in \mathbb{R}^{d_{\mathbb{H}}+1}$, and hyperspheres as $\mathbb{S}_{K_{\mathbb{S}}}^{d_{\mathbb{S}}} = \{\mathbf{x}_p \in \mathbb{R}^{d_{\mathbb{S}}+1} : \langle \mathbf{x}_p, \mathbf{x}_p \rangle_2 = 1/K_{\mathbb{S}}\}$, where $K_{\mathbb{S}} > 0$ and $\langle \cdot, \cdot \rangle_2$ is the standard Euclidean inner product $\langle \mathbf{x}, \mathbf{y} \rangle_2 = \sum_{j=1}^{d_{\mathbb{S}}+1} x_j y_j$, $\forall \mathbf{x}, \mathbf{y} \in \mathbb{R}^{d_{\mathbb{S}}+1}$. Table 1 provides a summary of relevant operators in Euclidean, hyperbolic, and spherical spaces with arbitrary curvatures. The closed forms for the distances between points in hyperbolic and spherical space use the arccosh and the arccos, their domains are $\{x \in \mathbb{R} : x \geqslant 1\}$ and $\{x \in \mathbb{R} : -1 \leqslant x \leqslant 1\}$, respectively. We apply clipping to avoid giving inputs close to the domain limits and prevent from instabilities during training.

Table 1: Relevant operators (exponential maps and distances between two points) in Euclidean, hyperbolic, and spherical spaces with arbitrary constant curvatures.

| Space Model | $exp_{\mathbf{x}_p}(\mathbf{x})$ | $\eth(\mathbf{x}, \mathbf{y})$ |
|---|---|---|
| $\mathbb{E}$, Euclidean | $\mathbf{x}_p + \mathbf{x}$ | $\|\|\mathbf{x} - \mathbf{y}\|\|_2$ |
| $\mathbb{H}$, hyperboloid | $\cosh\left(\sqrt{-K_{\mathbb{H}}}\|\|\mathbf{x}\|\|\right)\mathbf{x}_p + \sinh\left(\sqrt{-K_{\mathbb{H}}}\|\|\mathbf{x}\|\|\right)\frac{\mathbf{x}}{\sqrt{-K_{\mathbb{H}}}\|\|\mathbf{x}\|\|}$ | $\frac{1}{\sqrt{-K_{\mathbb{H}}}} \operatorname{arccosh}\left(K_{\mathbb{H}}\langle\mathbf{x}, \mathbf{y}\rangle_{\mathcal{L}}\right)$ |
| $\mathbb{S}$, hypersphere | $\cos\left(\sqrt{K_{\mathbb{S}}}\|\|\mathbf{x}\|\|\right)\mathbf{x}_p + \sin\left(\sqrt{K_{\mathbb{S}}}\|\|\mathbf{x}\|\|\right)\frac{\mathbf{x}}{\sqrt{K_{\mathbb{S}}}\|\|\mathbf{x}\|\|}$ | $\frac{1}{\sqrt{K_{\mathbb{S}}}} \arccos\left(K_{\mathbb{S}}\langle\mathbf{x}, \mathbf{y}\rangle_2\right)$ |

The latent output features produced by the neural network layers are in Euclidean space and must be mapped to the relevant model spaces before applying the distance metrics. We use the appropriate exponential map (refer to Table 1). To map Euclidean data to the hyperboloid, we use the hyperboloid north pole, that is, the origin $\mathbf{o}_{K_{\mathbb{H}}}^{\mathbb{H}} := (\frac{1}{\sqrt{-K_{\mathbb{H}}}}, 0, ..., 0) = (\frac{1}{\sqrt{-K_{\mathbb{H}}}}, \mathbf{0})$ as a reference point to perform tangent space operations. Using the trick described in Chami et al. (2019)[1], if $\mathbf{x}$ is an Euclidean feature we can consider $concat(0, \mathbf{x})$ to be a point in the manifold tangent space at $\mathbf{o}_{K_{\mathbb{H}}}^{\mathbb{H}}$. Therefore, we can obtain the mapped features $\overline{\mathbf{x}} = exp_{\mathbf{o}_{K_{\mathbb{H}}}^{\mathbb{H}}}^{\mathbb{H}}(concat(0, \mathbf{x}))$ via $\overline{\mathbf{x}} = \left(\frac{1}{\sqrt{-K_{\mathbb{H}}}}\cosh\left(\sqrt{-K_{\mathbb{H}}}\|\|\mathbf{x}\|\|\right), \sinh\left(\sqrt{-K_{\mathbb{H}}}\|\|\mathbf{x}\|\|\right)\frac{\mathbf{x}}{\sqrt{-K_{\mathbb{H}}}\|\|\mathbf{x}\|\|}\right)$. Similarly for the hypersphere using as reference point $\mathbf{o}_{K_{\mathbb{S}}}^{\mathbb{S}} := (\frac{1}{\sqrt{K_{\mathbb{S}}}}, 0, ..., 0)$, $\overline{\mathbf{x}} = \left(\frac{1}{\sqrt{K_{\mathbb{S}}}}\cos\left(\sqrt{K_{\mathbb{S}}}\|\|\mathbf{x}\|\|\right), \sin\left(\sqrt{K_{\mathbb{S}}}\|\|\mathbf{x}\|\|\right)\frac{\mathbf{x}}{\sqrt{K_{\mathbb{S}}}\|\|\mathbf{x}\|\|}\right)$.

## 3.2 PRODUCT MANIFOLDS

We define a product manifold as the Cartesian product $\mathcal{P} = \times_{i=1}^{n_{\mathcal{P}}} \mathcal{M}_{K_i}^{d_i}$, where $K_i$ and $d_i$ are the curvature and dimensionality of the manifold $\mathcal{M}_{K_i}^{d_i}$, respectively. We write points $\mathbf{x}_p \in \mathcal{P}$ using their coordinates $\mathbf{x}_p = concat\left(\mathbf{x}_p^{(1)}, \mathbf{x}_p^{(2)}, ..., \mathbf{x}_p^{(n_{\mathcal{P}})}\right) : \mathbf{x}_p^{(i)} \in \mathcal{M}_{K_i}^{d_i}$. Also, the metric of the product manifold decomposes into the sum of the constituent metrics $g_{\mathcal{P}} = \sum_{i=1}^{n_{\mathcal{P}}} g_i$, hence, $(\mathcal{P}, g_{\mathcal{P}})$ is also a Riemannian manifold if $(\mathcal{M}_{K_i}^{d_i}, g_i)$, $\forall i$ are all Riemannian manifolds in the first place. Note that the signature of the product space, that is, its parametrization, has several degrees of freedom: the number of components used, as well as the type of model spaces, their dimensionality, and curvature.

---

[1]Note that in this paper they define curvature differently.

If we restrict $\mathcal{P}$ to be composed of the Euclidean plane $\mathbb{E}_{K_\mathbb{E}}^{d_\mathbb{E}}$, hyperboloids $\mathbb{H}_{K_j^\mathbb{H}}^{d_j^\mathbb{H}}$, and hyperspheres $\mathbb{S}_{K_k^\mathbb{S}}^{d_k^\mathbb{S}}$ of constant curvature, we can write an arbitrary product manifold of model spaces as

$$\mathcal{P} = \mathbb{E}_{K_\mathbb{E}}^{d_\mathbb{E}} \times \left( \underset{j=1}{\overset{n_\mathbb{H}}{\times}} \mathbb{H}_{K_j^\mathbb{H}}^{d_j^\mathbb{H}} \right) \times \left( \underset{k=1}{\overset{n_\mathbb{S}}{\times}} \mathbb{S}_{K_k^\mathbb{S}}^{d_k^\mathbb{S}} \right) = \mathbb{E} \times \left( \underset{j=1}{\overset{n_\mathbb{H}}{\times}} \mathbb{H}_j \right) \times \left( \underset{k=1}{\overset{n_\mathbb{S}}{\times}} \mathbb{S}_k \right), \qquad (2)$$

where $K_\mathbb{E} = 0$, $K_j^\mathbb{H} < 0$, and $K_k^\mathbb{S} > 0$. The rightmost part of Equation 2 is included to simplify the notation. $\mathcal{P}$ would have a total of $1 + n_\mathbb{H} + n_\mathbb{S}$ component spaces, and total dimension $d_\mathbb{E} + \Sigma_{j=1}^{n_\mathbb{H}} d_j^\mathbb{H} + \Sigma_{k=1}^{n_\mathbb{S}} d_k^\mathbb{S}$. As shown in Gallier & Quaintance (2020), in the case of a product manifold as defined in Equation 2, the geodesics, exponential, and logarithmic maps on $\mathcal{P}$ are the concatenation of the corresponding notions of the individual model spaces.

## 3.3 DISTANCES AND SCALING METRICS FOR PRODUCT MANIFOLDS

To compute distances between points in the product manifold we can add up the square distances for the coordinates in each of the individual manifolds $\mathfrak{d}_\mathcal{P}(\overline{\mathbf{x}}_{p_1}, \overline{\mathbf{x}}_{p_2})^2 = \mathfrak{d}_\mathbb{E}\left(\mathbf{x}_{p_1}^{(1)}, \mathbf{x}_{p_2}^{(1)}\right)^2 + \Sigma_{j=1}^{n_\mathbb{H}} \mathfrak{d}_{\mathbb{H}_j}\left(\overline{\mathbf{x}}_{p_1}^{(1+j)}, \overline{\mathbf{x}}_{p_2}^{(1+j)}\right)^2 + \Sigma_{k=1}^{n_\mathbb{S}} \mathfrak{d}_{\mathbb{S}_k}\left(\overline{\mathbf{x}}_{p_1}^{(1+n_\mathbb{H}+k)}, \overline{\mathbf{x}}_{p_2}^{(1+n_\mathbb{H}+k)}\right)^2$, where the overline denotes that the adequate exponential map to project Euclidean feature entries to the relevant model space has been applied before computing the distance. In practice, this would be equivalent to mapping the feature outputs to the product manifold and operating on $\mathcal{P}$ directly. As suggested in Tabaghi et al. (2021), instead of directly updating the curvature of the hyperboloid and hypersphere model spaces used to construct the product manifold, we can set $K_j^\mathbb{H} = -1$, $\forall j$, and $K_k^\mathbb{S} = 1$, $\forall k$, and use a *scaled* distance metric instead. To do so, we introduce learnable coefficients $\alpha_j^\mathbb{H}$ and $\alpha_k^\mathbb{S}$,

$$\mathfrak{d}_\mathcal{P}(\overline{\mathbf{x}}_{p_1}, \overline{\mathbf{x}}_{p_2})^2 = \mathfrak{d}_\mathbb{E}\left(\mathbf{x}_{p_1}^{(1)}, \mathbf{x}_{p_2}^{(1)}\right)^2 + \Sigma_{j=1}^{n_\mathbb{H}}\left(\alpha_j^\mathbb{H} \mathfrak{d}_{\mathbb{H}_j}\left(\overline{\mathbf{x}}_{p_1}^{(1+j)}, \overline{\mathbf{x}}_{p_2}^{(1+j)}\right)\right)^2 + \Sigma_{k=1}^{n_\mathbb{S}}\left(\alpha_k^\mathbb{S} \mathfrak{d}_{\mathbb{S}_k}\left(\overline{\mathbf{x}}_{p_1}^{(1+n_\mathbb{H}+k)}, \overline{\mathbf{x}}_{p_2}^{(1+n_\mathbb{H}+k)}\right)\right)^2, \quad (3)$$

which is equivalent to learning the curvature of the non-Euclidean model spaces, but computationally more tractable and efficient (for further details on how this coefficients are updated refer to Appendix C.2). This newly defined distance metric can then be applied to calculate the probability of there existing an edge connecting latent features $p_{ij}^{(l)}(\boldsymbol{\Theta}^{(l)}) = \exp\left(-T\mathfrak{d}_\mathcal{P}\left(\overline{f_\boldsymbol{\Theta}^{(l)}(\mathbf{x}_i^{(l)})}, \overline{f_\boldsymbol{\Theta}^{(l)}(\mathbf{x}_j^{(l)})}\right)\right)$. Hence, $\mathcal{E}^{(l)}(\mathbf{X}^{(l)}; \boldsymbol{\Theta}^{(l)}, T, k, \mathfrak{d}_\mathcal{P}) = \{(i, j_{i,1}), (i, j_{i,2}), ..., (i, j_{i,k}) : i = 1, ..., N\}$. As discussed in Kazi et al. (2022), the logarithms of the edge probabilities are used to update the dDGM. This is done by incorporating an additional term to the network loss function which will be dependent on

$$\log p_{ij}^{(l)}(\boldsymbol{\Theta}^{(l)}) = -T\mathfrak{d}_\mathcal{P}\left(\overline{f_\boldsymbol{\Theta}^{(l)}(\mathbf{x}_i^{(l)})}, \overline{f_\boldsymbol{\Theta}^{(l)}(\mathbf{x}_j^{(l)})}\right) = -T\mathfrak{d}_\mathcal{P}\left(\overline{\mathbf{x}}_{p_i}^{(l)}, \overline{\mathbf{x}}_{p_j}^{(l)}\right) \qquad (4)$$

where the additional graph loss is given by $L_{GL} = \Sigma_{i=1}^N \left( \delta(y_i, \hat{y}_i) \Sigma_{l=1}^{l=L} \Sigma_{j:(i,j)\in\mathcal{E}^{(l)}} \log p_{ij}^{(l)} \right)$, and $\delta(y_i, \hat{y}_i) = \mathbb{E}(ac_i) - ac_i$ is a reward function based on the expected accuracy of the model. The loss function to update the dDGM model, $L_{GL}$, is identical to the original loss proposed by Kazi et al. (2022), for a brief review one may refer to Appendix C.1. Note that after passing the input $\mathbf{x}_i^{(l)}$ through the dDGM parameterized function $f_\boldsymbol{\Theta}^{(l)}$, the output $f_\boldsymbol{\Theta}^{(l)}(\mathbf{x}_i^{(l)}) = \hat{\mathbf{x}}_i^{(l+1)} = \mathbf{x}_{p_i}^{(l)}$ has dimension $d_\mathbb{E} + \Sigma_{j=1}^{n_\mathbb{H}} d_j^\mathbb{H} + \Sigma_{k=1}^{n_\mathbb{S}} d_k^\mathbb{S}$ and must be subdivided into $1 + n_\mathbb{H} + n_\mathbb{S}$ subarrays for each of the component spaces. Each subarray must be appropriately mapped to its model space. Hence, the overline in $\overline{f_\boldsymbol{\Theta}^{(l)}(\mathbf{x}_i^{(l)})}$ in Equation 4. Finally, Figure 1 summarizes the method described in this section. Note that $\mathbf{x}_p$ in Figure 1, would correspond to the concatenation of the origins of each model space composing the product manifold.

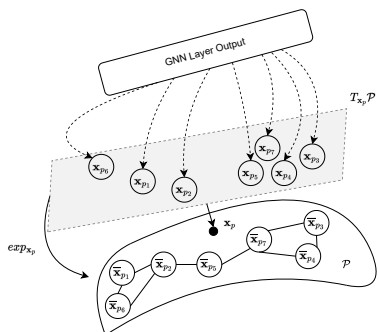

Figure 1: Diagram depicting mapping procedure from GNN Euclidean output to latent manifold $\mathcal{P}$. The appropriate exponential map is used to map the points from the tangent plane to the manifold. We construct the latent graph based on the distances on the learned manifold.

## 4 EXPERIMENTS AND RESULTS

The main objective of the experimental validation is to show that latent graph inference can benefit from using products of models spaces. To do so, we compare the performance of the dDGM module when using single model spaces against Cartesian products of model spaces. The model spaces are denoted as: Euclidean (dDGM-E/dDGM*-E, which is equivalent to the original architecture used by Kazi et al. (2022)), hyperbolic (dDGM-H/dDGM*-H), and spherical space (dDGM-S/dDGM*-S). The asterisk sign in the model name denotes that the dDGM* module is tasked with generating the latent graph without having access to the original adjacency matrix. dDGM models take as input $\mathbf{X}^{(0)}$ and $\mathbf{A}^{(0)}$, whereas dDGM* models only have access to $\mathbf{X}^{(0)}$. To refer to product manifolds, we simply append all the model spaces that compose the manifold to the name of the module, namely, the dDGM-SS module embedding space is a torus. In practice, we will use the same dimensionality but different curvature for each of the Cartesian components of the product manifolds. Lastly, if a GNN model uses the dDGM module, we name the diffusion layers and the latent graph inference module after. For example, GCN-dDGM-E refers to a GCN that instead of using the original dataset graph for diffusion, incorporates latent graph inference to the network and uses the Euclidean plane as embedding space.

Note that we only use a single latent graph inference module per neural network, that is, networks diffuse information based on only one latent graph. This is in line with previous work (Kazi et al. (2022)). Additionally, in Appendix E.1, we investigate the effect of leveraging multiple latent graphs in the same network and conclude that in general it is better to use a single latent graph due to computational efficiency and diminishing returns. The study regarding computational efficiency can be found in Appendix C.3. In particular, we compare the runtime speedup obtained using symbolic matrices as compared to standard dense PyTorch matrices. We observe that as more product manifolds and dDGM modules are included, the runtime speedup obtained using symbolic matrices becomes increasingly large. Moreover, without using symbolic matrices standard GPUs (we use NVIDIA P100 and Tesla T4) run out of memory for datasets with $\mathcal{O}(10^4)$ nodes such as PubMed, Physics, and CS. Hence, we recommend using symbolic matrices to help with scalability. Model architecture descriptions for all experiments can be found in Appendix G.

### 4.1 HOMOPHILIC AND HETEROPHILIC BENCHMARK GRAPH DATASETS

We first focus on standard graph datasets widely discussed in the Geometric Deep Learning literature such as Cora, CiteSeer (Yang et al. (2016); Lu & Getoor (2003); Sen et al. (2008)), PubMed, Physics and CS (Shchur et al. (2018)), which have high homophily levels ranging between 0.74 and 0.93. We also present the results for several heterophilic datasets, which have homophily levels between 0.11 and 0.23. In particular, we work with Texas, Wisconsin, Squirrel, and Chameleon (Rozemberczki et al., 2021). Results of particular interest for these datasets are recorded in Table 2 (benchmark models such as LDS-GNN (Franceschi et al., 2019), IDGL, IDGL-ANCH (Chen et al., 2020b), Pro-GNN, Pro-GNN-fs (Jin et al., 2020), and GCN-Jaccard (Wu et al., 2019) are also included). Additional experiments are available in Appendix E.2, E.3, and E.4, in which we perform an in depth

exploration of different hyperparameters, compare dDGMs and dDGM*s for all datasets, and try many product manifold combinations. Referring back to Table 2, we can see that product manifolds consistently outperform latent graph inference systems which only leverage a single model space for modeling the embedding space. Also note, that unlike in the work by Kazi et al. (2022), we do find single hyperbolic model spaces to often outperform inference systems that use the Euclidean plane as embbeding space. This shows that indeed mapping the Euclidean output features of the GNN layers to hyperbolic space using the exponential map before computing distances is of paramount importance (Kazi et al. (2022) ignored the exponential maps required to map features to the Poincaré ball).

Table 2: Results for heterophilic and homophilic datasets combining GCN diffusion layers with the latent graph inference system. We display results using model spaces as well as product manifolds to construct the latent graphs. The **First**, **Second** and **Third** best models for each dataset are highlighted in each table. $k$ denotes the number of connections per node when implementing the Gumbel Top-k sampling algorithm. Additional $k$ values are tested in Appendix E.2. Note that the models which use the Euclidean plane (former dDGM) as embedding space, denoted with an E in the table, are equivalent to those presented in Kazi et al. (2022).

| | HETEROPHILIC DATASETS | | | | | HOMOPHILIC DATASETS | | | | |
|---|---|---|---|---|---|---|---|---|---|---|
| | **Texas** | **Wisconsin** | **Squirrel** | **Chameleon** | | **Cora** | **CiteSeer** | **PubMed** | **Physics** | **CS** |
| Homophily level | 0.11 | 0.21 | 0.22 | 0.23 | Homophily level | 0.81 | 0.74 | 0.80 | 0.93 | 0.80 |
| Nodes | 183 | 251 | 5,201 | 2,277 | Nodes | 2,708 | 3,327 | 18,717 | 34,493 | 18,333 |
| Features | 1,703 | 1,703 | 2,089 | 2,325 | Features | 1,433 | 3,703 | 500 | 8,415 | 6,805 |
| Edges | 295 | 466 | 198,498 | 31,421 | Edges | 5,278 | 4,676 | 44,327 | 247,962 | 81,894 |
| Classes | 5 | 5 | 5 | 5 | Classes | 7 | 6 | 3 | 5 | 15 |
| Average Degree | 3.22 | 3.71 | 76.33 | 27.60 | Average Degree | 3.9 | 2.77 | 4.5 | 14.38 | 8.93 |
| **Former dDGM** | | | | | | | | | | |
| Model | Accuracy (%) $\pm$ Standard Deviation | | | | Model | Accuracy (%) $\pm$ Standard Deviation | | | | |
| $k$ | 2 | 10 | 3 | 5 | $k$ | 7 | 7 | 7 | 5 | 7 |
| GCN-dDGM*-E | $80.00_{\pm 8.31}$ | $88.00_{\pm 5.65}$ | $34.35_{\pm 2.34}$ | $48.90_{\pm 3.61}$ | GCN-dDGM-E | $82.11_{\pm 4.24}$ | $72.35_{\pm 1.92}$ | $87.69_{\pm 0.67}$ | $95.96_{\pm 0.40}$ | $87.17_{\pm 3.82}$ |
| **dDGM with Riemannian Geometry and Single Model Spaces (Ours)** | | | | | | | | | | |
| Model | Accuracy (%) $\pm$ Standard Deviation | | | | Model | Accuracy (%) $\pm$ Standard Deviation | | | | |
| $k$ | 2 | 10 | 3 | 5 | $k$ | 7 | 7 | 7 | 5 | 7 |
| GCN-dDGM*-H | $79.44_{\pm 7.88}$ | $89.03_{\pm 1.89}$ | $35.00_{\pm 2.35}$ | $48.28_{\pm 4.11}$ | GCN-dDGM-H | $84.68_{\pm 3.31}$ | $70.43_{\pm 4.95}$ | $87.74_{\pm 0.72}$ | $96.06_{\pm 0.42}$ | $88.78_{\pm 2.24}$ |
| GCN-dDGM*-S | $73.88_{\pm 9.95}$ | $85.33_{\pm 4.98}$ | $33.12_{\pm 2.22}$ | $48.63_{\pm 3.12}$ | GCN-dDGM-S | $80.44_{\pm 5.26}$ | $72.89_{\pm 2.00}$ | $87.13_{\pm 0.66}$ | $95.91_{\pm 0.41}$ | $84.16_{\pm 2.78}$ |
| **dDGM with Riemannian Geometry and Product Manifolds (Ours)** | | | | | | | | | | |
| Model | Accuracy (%) $\pm$ Standard Deviation | | | | Model | Accuracy (%) $\pm$ Standard Deviation | | | | |
| $k$ | 2 | 10 | 3 | 5 | $k$ | 7 | 7 | 7 | 10 | 3 |
| GCN-dDGM*-HH | $78.89_{\pm 8.53}$ | $88.00_{\pm 3.26}$ | $34.38_{\pm 1.07}$ | $48.33_{\pm 4.14}$ | GCN-dDGM-HH | $76.09_{\pm 7.11}$ | $71.27_{\pm 2.09}$ | $87.50_{\pm 0.91}$ | $94.73_{\pm 2.83}$ | $82.91_{\pm 3.00}$ |
| GCN-dDGM*-SS | $73.89_{\pm 8.62}$ | $74.66_{\pm 18.85}$ | $34.06_{\pm 2.20}$ | $48.28_{\pm 3.07}$ | GCN-dDGM-SS | $65.96_{\pm 9.46}$ | $59.16_{\pm 5.96}$ | $87.82_{\pm 0.59}$ | $90.72_{\pm 5.26}$ | $59.31_{\pm 7.18}$ |
| GCN-dDGM*-EH | $81.67_{\pm 7.05}$ | $86.67_{\pm 3.77}$ | $34.37_{\pm 1.72}$ | $47.58_{\pm 3.85}$ | GCN-dDGM-EH | $82.32_{\pm 4.71}$ | $72.89_{\pm 1.64}$ | $87.41_{\pm 0.80}$ | $96.03_{\pm 0.37}$ | $91.37_{\pm 1.28}$ |
| GCN-dDGM*-ES | $81.11_{\pm 10.30}$ | $76.00_{\pm 11.31}$ | $33.38_{\pm 1.86}$ | $47.49_{\pm 3.60}$ | GCN-dDGM-ES | $81.44_{\pm 5.80}$ | $71.87_{\pm 3.20}$ | $87.50_{\pm 0.65}$ | $95.41_{\pm 1.73}$ | $90.87_{\pm 0.82}$ |
| GCN-dDGM*-HS | $81.11_{\pm 9.69}$ | $86.67_{\pm 1.89}$ | $34.65_{\pm 2.45}$ | $47.84_{\pm 2.67}$ | GCN-dDGM-HS | $82.59_{\pm 4.50}$ | $72.77_{\pm 2.76}$ | $85.89_{\pm 4.29}$ | $95.84_{\pm 0.29}$ | $89.43_{\pm 2.37}$ |
| GCN-dDGM*-EHH | $81.11_{\pm 5.09}$ | $77.60_{\pm 8.62}$ | $33.19_{\pm 1.92}$ | $44.27_{\pm 2.96}$ | GCN-dDGM-EHH | $86.63_{\pm 3.25}$ | $75.42_{\pm 2.39}$ | $39.93_{\pm 1.35}$ | $95.63_{\pm 1.36}$ | $92.86_{\pm 0.96}$ |
| GCN-dDGM*-EHS | $79.44_{\pm 6.11}$ | $89.33_{\pm 1.89}$ | $34.17_{\pm 2.23}$ | $47.58_{\pm 4.54}$ | GCN-dDGM-EHS | $83.58_{\pm 4.39}$ | $69.98_{\pm 2.70}$ | $87.05_{\pm 1.38}$ | $96.21_{\pm 0.44}$ | $89.93_{\pm 3.86}$ |
| **Vanilla Architectures** | | | | | | | | | | |
| Model | Accuracy (%) $\pm$ Standard Deviation | | | | Model | Accuracy (%) $\pm$ Standard Deviation | | | | |
| MLP* | $77.78_{\pm 10.24}$ | $85.33_{\pm 4.99}$ | $30.44_{\pm 2.55}$ | $40.35_{\pm 3.37}$ | MLP* | $58.92_{\pm 3.28}$ | $59.48_{\pm 2.14}$ | $85.75_{\pm 1.02}$ | $94.91_{\pm 0.30}$ | $87.80_{\pm 1.54}$ |
| GCN | $41.66_{\pm 11.72}$ | $47.20_{\pm 9.76}$ | $24.19_{\pm 2.56}$ | $32.56_{\pm 3.53}$ | GCN | $83.11_{\pm 2.29}$ | $69.97_{\pm 2.06}$ | $85.75_{\pm 1.01}$ | $95.51_{\pm 0.34}$ | $87.28_{\pm 1.54}$ |

We have shown that the latent graph inference system enables GCN diffusion layers to achieve good performance for heterophilic datasets. We hypothesize that for this to be possible, it should be able to generate homophilic latent graphs, on which GCNs can easily diffuse. In Table 3 we display the homophily levels of the learned latent graphs, which corroborates our intuition. As we can see from the results all models are able to generate latent graphs with higher homophily than those of the original dataset graphs. The latent graph inference system seems to find it easier to increase the homophily levels of smaller datasets, which is reasonable since there is less information to reorganize. There is a clear correlation between model performance in terms of accuracy (Table 2) and the homophily level that the dDGM* modules are able to achieve for the latent graphs (Table 3).

Table 3: Homophily level of the learned latent graphs. Latent graph inference modules which use different manifolds to generate their respective latent graphs achieve different homophily levels. Also, depending on weight initialization, the inference system can converge to slightly different latent graphs.

| | **Texas** | **Wisconsin** | **Squirrel** | **Chameleon** |
|---|---|---|---|---|
| Original Graph Homophily | 0.11 | 0.21 | 0.22 | 0.23 |
| $k$ | 2 | 10 | 3 | 5 |
| Model | Latent Graph Homophily $h$ $\pm$ Standard Deviation | | | |
| GCN-dDGM*-E | $0.89_{\pm 0.02}$ | $0.69_{\pm 0.01}$ | $0.33_{\pm 0.00}$ | $0.37_{\pm 0.02}$ |
| GCN-dDGM*-H | $0.89_{\pm 0.01}$ | $0.66_{\pm 0.01}$ | $0.33_{\pm 0.00}$ | $0.46_{\pm 0.05}$ |
| GCN-dDGM*-S | $0.86_{\pm 0.03}$ | $0.64_{\pm 0.01}$ | $0.32_{\pm 0.00}$ | $0.45_{\pm 0.04}$ |
| GCN-dDGM*-HH | $0.91_{\pm 0.01}$ | $0.66_{\pm 0.02}$ | $0.32_{\pm 0.00}$ | $0.37_{\pm 0.01}$ |
| GCN-dDGM*-SS | $0.85_{\pm 0.02}$ | $0.51_{\pm 0.01}$ | $0.27_{\pm 0.00}$ | $0.31_{\pm 0.01}$ |
| GCN-dDGM*-EH | $0.90_{\pm 0.01}$ | $0.65_{\pm 0.01}$ | $0.32_{\pm 0.00}$ | $0.43_{\pm 0.03}$ |
| GCN-dDGM*-ES | $0.91_{\pm 0.00}$ | $0.63_{\pm 0.02}$ | $0.27_{\pm 0.02}$ | $0.32_{\pm 0.04}$ |
| GCN-dDGM*-HS | $0.91_{\pm 0.02}$ | $0.67_{\pm 0.02}$ | $0.33_{\pm 0.00}$ | $0.36_{\pm 0.08}$ |
| GCN-dDGM*-EHH | $0.90_{\pm 0.05}$ | $0.59_{\pm 0.04}$ | $0.43_{\pm 0.03}$ | $0.55_{\pm 0.03}$ |
| GCN-dDGM*-EHS | $0.91_{\pm 0.03}$ | $0.66_{\pm 0.03}$ | $0.32_{\pm 0.01}$ | $0.45_{\pm 0.01}$ |

For example, in the case of Texas we achieve the highest homophily levels for the latent graphs of between $0.85 \pm 0.02$ and $0.91 \pm 0.03$, and also some of the highest accuracies ranging from $73.88 \pm 9.95\%$ to $81.67 \pm 7.05\%$. For Wisconsin the homophily level is lower than that for Texas, but this can be attributed to the fact that $k = 10$, inevitably creating more connections with nodes from other classes. Also, in Wisconsin there are two classes with substantially less nodes than the rest, meaning that a high accuracy can be achieved even if those are misclassified. On the other hand, for Squirrel, although the latent graph inference system still manages to increase homophily from $0.22$ in the original graph to between $0.27 \pm 0.00$ and $0.43 \pm 0.03$ in the latent graph, the increase is not big as compared to the other datasets and we can see how this also has an effect on performance. In Table 2 the maximum accuracy for Squirrel is of $35.00 \pm 2.35\%$. Note that this is still substantially better than using a MLP or a standard GCN, which obtain accuracies of $30.44 \pm 2.55\%$ and $24.19 \pm 2.56\%$, respectively. The same discussion applies to Chameleon. Figure 2 displays how the graph connectivity is modified during the training process. This shows that the inference system is able to dynamically learn an optimal connectivity structure for the latent graph based on the downstream task, and modify it accordingly during training. Additional latent graph plots for the different datasets can be found in Appendix F.

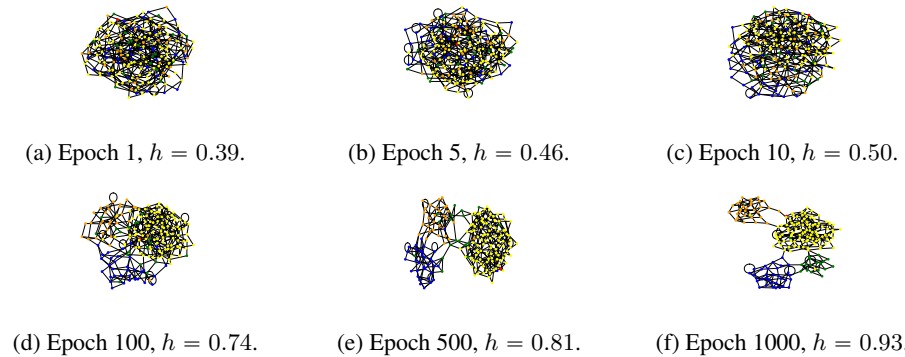

(a) Epoch 1, $h = 0.39$.      (b) Epoch 5, $h = 0.46$.      (c) Epoch 10, $h = 0.50$.

(d) Epoch 100, $h = 0.74$.      (e) Epoch 500, $h = 0.81$.      (f) Epoch 1000, $h = 0.93$.

Figure 2: Latent graph homophily level, $h$, evolution as a function of training epochs for Texas. The latent graphs are produced during the training process for the GCN-dDGM*-EH model with $k = 2$.

## 4.2 Real-World Applications

Next, we test the latent graph inference system on real-world applications: on the TadPole dataset (Marinescu et al. (2020)), which was also discussed by Kazi et al. (2022), and the Aerothermodynamics dataset, which we have created for this work. The TadPole dataset contains information about brain images for different patients and the task is to classify each patient into three classes: *Normal Control*, *Alzheimer's Disease* and *Mild Cognitive Impairment*. On the other hand, the Aerothermodynamics dataset was inspired by recent research discussing potential applications of machine learning to aerospace engineering (Maheshwari et al. (2018); Sáez de Ocáriz Borde et al. (2021); de Ocáriz Borde et al. (2021)), and challenges networks to classify different regions of a shock wave around a rocket (refer to Appendix D for more information on the dataset). Note that since neither of these datasets have a graph structure (they are pointclouds), only the dDGM* can be utilized in this case. Results are given in Table 4. We use GAT diffusion layers and compare the performance using single model spaces and product manifolds. Almost all models using the latent graph inference system outperform the performance of the MLP. It is important to note that since the datasets do not provide an input graph it would not be possible to use GAT models without using dDGM* modules. Again, we find that using product manifolds to model the latent space of potentially complex real-world datasets proves beneficial and boosts model accuracy.

In principle, the Aerothermodynamics datasets classifies shock regions based on the flow absolute velocity into 4 regions as recorded in Table 4. However, we tested increasing the number of classes by further subdividing the flow into a total of 7 regions, as shown in Figure 3. We found that interestingly, the latent graph inferred by the model does not only cluster nodes with similar labels together, but it actually organizes the latent graph in order based on the absolute velocities (which are not explicitly given to the model: the velocities are used to create the labels but values are

not provided to the model as input). This suggests that the graph generation system is organizing the latent graph based on some inferred high level understanding of the physics of the problem. Additional latent graphs for these datasets are provided in Appendix F.2.

| | TadPole | Aerothermodynamics |
|---|---|---|
| Nodes | 564 | 1,456 |
| Features | 30 | 1 |
| Classes | 3 | 4 |
| Model | Accuracy (%) $\pm$ Standard Deviation | |
| GAT-dDGM*-E | $90.36_{\pm 3.21}$ | $89.20_{\pm 1.81}$ |
| GAT-dDGM*-H | $88.75_{\pm 3.91}$ | $86.36_{\pm 2.99}$ |
| GAT-dDGM*-S | $90.89_{\pm 4.55}$ | $86.21_{\pm 5.37}$ |
| GAT-dDGM*-HH | $89.68_{\pm 5.70}$ | $77.47_{\pm 13.81}$ |
| GAT-dDGM*-SS | $89.82_{\pm 4.79}$ | $71.26_{\pm 14.40}$ |
| GAT-dDGM*-EH | $90.36_{\pm 4.16}$ | $89.90_{\pm 0.55}$ |
| GAT-dDGM*-ES | $89.46_{\pm 5.56}$ | $90.34_{\pm 3.90}$ |
| GAT-dDGM*-HS | $86.43_{\pm 5.82}$ | $88.73_{\pm 4.72}$ |
| GAT-dDGM*-EHH | $87.68_{\pm 9.95}$ | $91.72_{\pm 0.98}$ |
| GAT-dDGM*-EHS | $92.68_{\pm 3.52}$ | $88.74_{\pm 3.39}$ |
| MLP* | $87.68_{\pm 3.52}$ | $81.03_{\pm 8.56}$ |

Table 4: Results for the TadPole and the Aerothermodynamics datasets using GAT diffusion layers and different latent graph inference modules.

Figure 3: Latent graph obtained by the GAT-dDGM*-EHH model with $k = 7$ including more subclasses for different absolute velocity simulation regions for the Aerothermodynamics dataset.

### 4.3 SCALING TO LARGE GRAPHS

All datasets considered so far are relatively small. In this section we work with datasets from the Open Graph Benchmark (OGB) which contain large-scale graphs and require models to perform realistic out-of-distribution generalization. In particular, we use the OGB-Arxiv and the OGB-Products datasets. As discussed in Appendix C.3.3, for training on these datasets we use graph subsampling techniques, and Graph Attention Network version 2 (GATv2) diffusion layers (Brody et al. (2021)); since we do not expect overfitting we can use more expressive layers. OGB-Arxiv and OGB-Products have a total of 40 and 47 classes, respectively. Previous datasets only considered multi-class classification for between 3 to 15 classes. This, added to the fact that the datasets have orders of magnitude more nodes and edges, makes the problems in this section considerably more challenging. For OGB-Arxiv using a MLP and a GATv2 model without leveraging latent graph inference we obtain accuracies of $63.49 \pm 0.15\%$ and $61.93 \pm 1.62\%$, respectively. The best model with latent graph inference, GATv2-dDGM*-EHS, achieves an accurary of $65.06 \pm 0.09\%$. For OGB-Products, the MLP and a GATv2 results are $66.05 \pm 0.20\%$ and $62.02 \pm 2.60\%$, and for the best model, GATv2-dDGM-E, we record an accuracy of $66.59 \pm 0.30\%$. From the results (more in Appendix E.5), we conclude that latent graph inference is still beneficial for this larger datasets but there is substantial room for improvement. Graph subsampling interferes with embedding space learning.

## 5 DISCUSSION AND CONCLUSION

In this work we have incorporated Riemannian geometry to the dDGM latent graph inference module by Kazi et al. (2022). First, we have shown how to work with manifolds of constant arbitrary curvature, both positive and negative. Next, we have leveraged product manifolds of model spaces and their convenient mathematical properties to enable the dDGM module to generate a more complex homogeneous manifold with varying curvature which can better encode the latent data, while learning the curvature of each model space composing the product manifold during training.

We have evaluated our method on many and diverse datasets, and we have shown that using product manifolds to model the embedding space for the latent graph gives enhanced downstream performance as compared to using single model spaces of constant curvature. The inference system has been tested on both homophilic and heterophilic benchmarks. In particular, we have found that using optimized latent graphs, diffusion layers like GCNs are able to successfully operate on datasets with low homophily levels. Additionally, we have tested and proven the applicability of our method to large-scale graphs. Lastly, we have shown the benefits of applying this procedure in real-world problems such as brain imaging based data and aerospace engineering problems. All experiments

discussed in the main text are concerned with transductive learning; however, the method is also applicable to inductive learning, see Appendix E.6.

The product manifold embedding space approach has provided a computationally tractable way of generating more complex homogenous manifolds for the latent features' embedding space. Furthermore, the curvature of the product components is learned rather than it being a fixed hyperparameter, which allows for greater flexibility. However, the number of model spaces to generate the product manifold must be specified before training. It would be interesting to devise an approach for the network to independently add more model spaces to the product manifold when needed. Also, we are restricting our approach to product manifolds based on model spaces of constant curvature due to their suitable mathematical properties. Such product manifolds do not cover all possible arbitrary manifolds in which the latent data could be encoded and hence, there could still be, mathematically speaking, more optimal manifolds to represent the data. It is worth exploring whether approaches to generate even more diverse and computationally tractable manifolds would be possible.

**Future Work** Lastly, there are a few limitations intrinsic to the dDGM module, irrespective of the product manifold embedding approach introduced in this work. Firstly, although utilizing symbolic matrices can help computational efficiency (Appendix C.3), the method still has quadratic complexity. Kazi et al. (2022) proposed computing probabilities in a neighborhood of the node and using tree-based algorithms to reduce it to $\mathcal{O}(n \log n)$. Moreover, the Gumbel Top-k sampling approach restricts the average node degree of the latent graph and requires manually adjusting the $k$ value through testing. A possible solution could be to use a distance based sparse threshold approach in which an unweighted edge is created between two nodes if they are within a threshold distance of each other in latent space. This is similar to the Gumbel Top-k trick, but instead of choosing a fixed number of closest neighbors, we connect all nodes within a distance. This could help better capture the heterogeneity of the graph. However, we actually tested this approach and found it quite unstable. Note that although we do not have the $k$ parameter anymore, we must still choose a threshold distance. Another avenue to help with scalability, improve computational complexity, and facilitate working with large-scale graphs would be to use a hierarchical perspective. Inspired by brain interneurons (Freund & Buzsáki (1996)), we could introduce fictitious connector inducing nodes in different regions of the graph, use those nodes to summarize different regions of large graphs, and apply the kNN algorithm or the Gumbel Top-k trick to the fictitious connector inducing nodes. This way the computational complexity would still be quadratic, but proportional to the number of interconnectors. Similar techniques have been applied to Gaussian Processes (Galy-Fajou & Opper (2021); Wu et al. (2021)) and Set Transformers (Lee et al. (2019)).

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

## A   RIEMANNIAN MANIFOLDS

In Riemannian geometry, we define a Riemannian manifold or Riemannian space $(\mathcal{M}, g)$ as a real and differentiable manifold $\mathcal{M}$ in which each tangent space has an associated inner product $g$, that is, a Riemannian metric, which must vary smoothly when considering points in the manifold. The Riemannian manifold $\mathcal{M} \subseteq \mathbb{R}^N$ (lives in the ambient space $\mathbb{R}^N$) is a collection of real vectors, and is locally similar to a linear space. The Riemannian metric generalizes inner products for Riemannian manifolds. It also allows to define geometric notions on a Riemannian manifold such as lengths of curves, curvature, and angles to name a few. If the point $\mathbf{x}_p \in \mathcal{M}$, then we can denote the tangent space at $\mathbf{x}_p$ as $T_{\mathbf{x}_p}\mathcal{M}$, which has the same dimensionality as $\mathcal{M}$. $T_{\mathbf{x}_p}\mathcal{M}$ is the collection of all tangent vectors at $\mathbf{x}_p$. Moreover, $g_{\mathbf{x}_p} : T_{\mathbf{x}_p}\mathcal{M} \times T_{\mathbf{x}_p}\mathcal{M} \to \mathbb{R}$ is given by a positive-definite inner product in the tangent space and depends smoothly on $\mathbf{x}_p$. A geodesic represents the shortest smooth path between two points in a Riemannian manifold and generalizes the notion of a straight line in Euclidean space. The length of a smooth continuously differentiable curve $\gamma : t \to \gamma(t) \in \mathcal{M}, t \in [0, 1]$ is given by

$$L(\gamma) = \int_0^1 ||\gamma'(t)|| \, dt. \tag{5}$$

Note that $L(\gamma)$ is unchanged by a monotone reparametrization. The geodesic distance between two points $\mathbf{x}_{p_i}, \mathbf{x}_{p_j} \in \mathcal{M}$ is defined as the infimum (greatest lower bound) of the length taken over all piecewise continuously differentiable curves, such that

$$\gamma_{\mathbf{x}_{p_i}, \mathbf{x}_{p_j}} = \underset{\gamma}{\mathrm{argmin}} \, L(\gamma) : \gamma(0) = \mathbf{x}_{p_i}, \gamma(1) = \mathbf{x}_{p_j}. \tag{6}$$

The norm of a tangent vector $\mathbf{v} \in T_{\mathbf{x}_p}\mathcal{M}$ is given by

$$||\mathbf{v}|| = \sqrt{g_{\mathbf{x}_p}(\mathbf{v}, \mathbf{v})}. \tag{7}$$

Moving from a point $\mathbf{x}_p \in \mathcal{M}$ with initial constant velocity $\mathbf{v} \in T_{\mathbf{x}_p}\mathcal{M}$ is formalized by the exponential map

$$exp_{\mathbf{x}_p} : T_{\mathbf{x}_p}\mathcal{M} \to \mathcal{M}, \tag{8}$$

which gives the position of the geodesic at $t = 1$ so that

$$exp_{\mathbf{x}_p}(\mathbf{v}) = \gamma(1). \tag{9}$$

There is a unique unit speed geodesic $\gamma$ which satisfies $\gamma(0) = \mathbf{x}_p$ and $\gamma'(0) = \mathbf{v}$. On the other hand, and less relevant to the work at hand, the logarithmic map is the inverse of the exponential map

$$log_{\mathbf{x}_p} = exp_{\mathbf{x}_p}^{-1} : \mathcal{M} \to T_{\mathbf{x}_p}\mathcal{M}. \tag{10}$$

In geodesically complete Riemannian manifolds, both the exponential and logarithm maps are well-defined (Needham (1997)).

In general, it can be impossible to find a solution for the geodesic between two points for an arbitrary manifold. As discussed in the main text, we want to be able to move beyond constant curvature spaces and compute the similarity measure $\varphi$ for latent features which may reside in a more general and learnable manifold. This will enable us to more accurately model data which may comprise varying structure, beyond that which can be represented in Euclidean space, and also beyond hierarchical or cyclical data which can be linked to constant curvature hyperbolic and spherical spaces, respectively. That is, by varying structure we refer to data which may present different underlying patterns in different regions of space. Generating more complex manifolds we intend to minimize the distortion incurred in the data and to represent the points in a more suitable manifold for the downstream task. However, we must still be able to map the Euclidean output features of the network to our learnable manifold and to compute distances between points. To generate a more complex and learnable manifold while still having a closed form solution for the exponential map and geodesics, we will introduce a product manifold embedding space composed of multiple copies of simple model spaces with constant curvature. Although product manifolds based on constant curvature models still fall under the homogeneous manifold category (Kowalski et al. (1989)), they allow to model more complex embedding spaces for the dDGM module than those that can be represented using only constant curvature homogeneous spaces.

For example, the standard torus, which can be obtained by multiplying two spheres, is a homogeneous manifold. Nevertheless, some points on the manifold have positive Gaussian curvature (outer part of the torus, elliptic points), some have negative (inner part, hyperbolic points), and others have zero (parabolic points). This is because for the torus the Euler characteristic (Beltramo et al. (2021)) is zero so it must always have regions of negative Gaussian curvature. This is to counterbalance the regions of positive curvature guaranteed in Hilbert's theorem. The main point is that using product manifolds of constant curvature spaces we can generate manifolds with regions with different Gaussian curvature that can help us better represent data structures which may not be purely Euclidean, hyperbolic, or spherical.

For all points on the manifold, we can define a normal vector that is at right angles to the surface. By generating intersections between normal planes (that contain the normal vector) and the surface we can compute normal sections. In general, different normal sections will have different curvatures. We refer to $\kappa_1$ and $\kappa_2$ as the principal curvatures. They correspond to the maximum and minimum values that the curvature can take at a given point (Kühnel (2005)). The Gaussian curvature $K$ is the product of the two

$$K = \kappa_1\kappa_2. \tag{11}$$

A Riemannian manifold is said to have constant Gaussian curvature $K$ if $sec(P) = K$ for all two-dimensional linear subspaces $P \subset T_{\mathbf{x}_p}\mathcal{M}$ and for all $\mathbf{x}_p \in \mathcal{M}$. Manifolds can be classified into three classes depending on their curvature: flat space, positively curved space, and negatively curved space.

## B FURTHER DISCUSSION ON CURVATURE LEARNING

Next, we aim to provide a more detailed explanation on the topic of curvature learning for product manifolds and the implemented procedure using learnable distance-metric-scaling coefficients. Let us discuss product manifolds which are solely generated based on Cartesian products of the same model spaces, such as

$$
\mathcal{P}_{\mathbb{H}} = \bigtimes_{j=1}^{n_{\mathbb{H}}} \mathbb{H}_{K_j^{\mathbb{H}}}^{d_j^{\mathbb{H}}},
\tag{12}
$$

which uses Cartesian products of hyperboloids. Likewise, taking Cartesian products of hyperspheres we would obtain

$$
\mathcal{P}_{\mathbb{S}} = \bigtimes_{k=1}^{n_{\mathbb{S}}} \mathbb{S}_{K_k^{\mathbb{S}}}^{d_k^{\mathbb{S}}}.
\tag{13}
$$

For these cases, although $\mathfrak{d}_{\mathbb{H}_{K_j^{\mathbb{H}}}^{d_j^{\mathbb{H}}}}\left(\overline{\mathbf{x}}_{p_1}^{(j)}, \overline{\mathbf{x}}_{p_2}^{(j)}\right)$ and $\mathfrak{d}_{\mathbb{S}_{K_k^{\mathbb{S}}}^{d_k^{\mathbb{S}}}}\left(\overline{\mathbf{x}}_{p_1}^{(k)}, \overline{\mathbf{x}}_{p_2}^{(k)}\right)$ are in practice computed all using hyperboloids and hyperspheres with the same fixed curvature $K_j^{\mathbb{H}} = -1$, $\forall j$, and $K_k^{\mathbb{S}} = 1$, $\forall k$, the scaling coefficients control the curvature of the spaces individually,

$$
\mathfrak{d}_{\mathcal{P}_{\mathbb{H}}}(\overline{\mathbf{x}}_{p_1}, \overline{\mathbf{x}}_{p_2}) = \sqrt{\sum_{j=1}^{n_{\mathbb{H}}} \left(\alpha_j^{\mathbb{H}} \mathfrak{d}_{\mathbb{H}_{K_j^{\mathbb{H}}}^{d_j^{\mathbb{H}}}}\left(\overline{\mathbf{x}}_{p_1}^{(j)}, \overline{\mathbf{x}}_{p_2}^{(j)}\right)\right)^2},
\tag{14}
$$

and,

$$
\mathfrak{d}_{\mathcal{P}_{\mathbb{S}}}(\overline{\mathbf{x}}_{p_1}, \overline{\mathbf{x}}_{p_2}) = \sqrt{\sum_{k=1}^{n_{\mathbb{S}}} \left(\alpha_k^{\mathbb{S}} \mathfrak{d}_{\mathbb{S}_{K_k^{\mathbb{S}}}^{d_k^{\mathbb{S}}}}\left(\overline{\mathbf{x}}_{p_1}^{(k)}, \overline{\mathbf{x}}_{p_2}^{(k)}\right)\right)^2}.
\tag{15}
$$

That is, $\alpha_j^{\mathbb{H}}$ and $\alpha_k^{\mathbb{S}}$ scale the distances generated based on unit hyperboloids and hyperspheres, respectively. Using the scaled metrics, we are effectively still computing Cartesian products of hyperboloids and hyperspheres with different curvatures, so that

$$
\mathcal{P}_{\mathbb{H}} = \bigtimes_{j=1}^{n_{\mathbb{H}}} \mathbb{H}_{K_j^{\mathbb{H}}}^{d_j^{\mathbb{H}}} \neq \bigtimes_{j=1}^{n_{\mathbb{H}}} \mathbb{H}_{K_{\mathbb{H}}}^{d_j^{\mathbb{H}}},
\tag{16}
$$

$$
\mathcal{P}_{\mathbb{S}} = \bigtimes_{k=1}^{n_{\mathbb{S}}} \mathbb{S}_{K_k^{\mathbb{S}}}^{d_k^{\mathbb{S}}} \neq \bigtimes_{k=1}^{n_{\mathbb{S}}} \mathbb{S}_{K_{\mathbb{S}}}^{d_k^{\mathbb{S}}}.
\tag{17}
$$

This guarantees we are retrieving equivalent values to those which would be generated by model spaces with different curvatures. This is done to avoid backpropagating through operators such as exponential maps and closed form solutions for distances in the different model spaces.

Considering again an arbitrary product manifold of all model spaces as in Equation 2, we can control the curvature through the following derivatives

$$
\frac{\partial \mathfrak{d}_{\mathcal{P}}}{\partial \alpha_j^{\mathbb{H}}},
\tag{18}
$$

for the hyperboloid terms, and

$$\frac{\partial \mathfrak{d}_{\mathcal{P}}}{\partial \alpha_k^{\mathbb{S}}}, \tag{19}$$

for the hyperspheres. In the case of Euclidean space the curvature is always $K_{\mathbb{E}} = 0$, so there is no need to learn it. Also, using a Cartesian product of Euclidean spaces would be equivalent to using a single Euclidean space of greater dimensionality

$$\mathcal{P}_{\mathbb{E}} = \mathop{\vcenter{\hbox{$\times$}}}_{i=1}^{n_{\mathbb{E}}} \mathbb{E}_{K_i^{\mathbb{E}}}^{d_i^{\mathbb{E}}} = \mathbb{E}_{K_{\mathbb{E}}}^{d_{\mathbb{E}}}, \tag{20}$$

where $K_i^{\mathbb{E}} = K_{\mathbb{E}} = 0$, $\forall i$, and $d_{\mathbb{E}} = \sum_i^{n_{\mathbb{E}}} d_i^{\mathbb{E}}$. Hence, in Equation 2 we used a single Euclidean space. The behavior described in Equation 20 can be better appreciated by comparing distance $\mathfrak{d}_{\mathcal{P}_{\mathbb{E}}}$ obtained using $\mathcal{P}_{\mathbb{E}}$ and $\mathfrak{d}_{\mathbb{E}_{K_{\mathbb{E}}}^{d_{\mathbb{E}}}}$ from $\mathbb{E}_{K_{\mathbb{E}}}^{d_{\mathbb{E}}}$:

$$\mathfrak{d}_{\mathcal{P}_{\mathbb{E}}}^2 = \sum_{i=1}^{n_{\mathbb{E}}} \left( \mathfrak{d}_{\mathbb{E}_{K_i^{\mathbb{E}}}^{d_i^{\mathbb{E}}}} \right)^2 = \sum_{i=1}^{n_{\mathbb{E}}} \left( \mathfrak{d}_{\mathbb{E}_{K_{\mathbb{E}}}^{d_i^{\mathbb{E}}}} \right)^2, \tag{21}$$

$$\mathfrak{d}_{\mathcal{P}_{\mathbb{E}}}^2 = \mathfrak{d}_{\mathbb{E}_{K_{\mathbb{E}}}^{d_1^{\mathbb{E}}}}^2 + \mathfrak{d}_{\mathbb{E}_{K_{\mathbb{E}}}^{d_2^{\mathbb{E}}}}^2 + ... + \mathfrak{d}_{\mathbb{E}_{K_{\mathbb{E}}}^{d_{n_{\mathbb{E}}}^{\mathbb{E}}}}^2. \tag{22}$$

Considering that the equation for independent distances in each Euclidean space is

$$\mathfrak{d}_{\mathbb{E}_{K_{\mathbb{E}}}^{d_i^{\mathbb{E}}}} = \sqrt{\sum_a \left( \mathbf{x}_{p_1}^{(a)} - \mathbf{x}_{p_2}^{(a)} \right)^2}, \tag{23}$$

we obtain,

$$\mathfrak{d}_{\mathcal{P}_{\mathbb{E}}} = \sqrt{\sum_a \left( \mathbf{x}_{p_1}^{(a)} - \mathbf{x}_{p_2}^{(a)} \right)^2 + \sum_b \left( \mathbf{x}_{p_1}^{(b)} - \mathbf{x}_{p_2}^{(b)} \right)^2 + ... + \sum_z \left( \mathbf{x}_{p_1}^{(z)} - \mathbf{x}_{p_2}^{(z)} \right)^2}, \tag{24}$$

which is analogous to taking the second power of the difference of two points in another Euclidean space with more dimensions, so that

$$\mathfrak{d}_{\mathcal{P}_{\mathbb{E}}} = \mathfrak{d}_{\mathbb{E}_{K_{\mathbb{E}}}^{d_{\mathbb{E}}}}. \tag{25}$$

This behavior is only applicable to Euclidean space (Gu et al. (2019)), when multiplying other model spaces

$$\mathcal{P}_{\mathbb{H}} = \mathop{\vcenter{\hbox{$\times$}}}_{j=1}^{n_{\mathbb{H}}} \mathbb{H}_{K_j^{\mathbb{H}}}^{d_j^{\mathbb{H}}} \neq \mathbb{H}_{K_{\mathbb{H}}}^{d_{\mathbb{H}}}, \tag{26}$$

$$\mathcal{P}_{\mathbb{S}} = \mathop{\vcenter{\hbox{$\times$}}}_{k=1}^{n_{\mathbb{S}}} \mathbb{S}_{K_k^{\mathbb{S}}}^{d_k^{\mathbb{S}}} \neq \mathbb{S}_{K_{\mathbb{S}}}^{d_{\mathbb{S}}}, \tag{27}$$

even when the curvature is the same for all hyperboloids and hyperspheres, $K_j^{\mathbb{H}} = K_{\mathbb{H}}$, $\forall j$ and $K_k^{\mathbb{S}} = K_{\mathbb{S}}$, $\forall k$. For example, multiplying two hyperspheres will result in a hypertorus, not a higher-dimensional hypersphere.

As a final remark, note that any loss function which is dependent on the distance function induced by the Riemannian metric of a Riemannian manifold is locally smooth on $\mathcal{M}$ (where $\mathcal{M}$ could be a product manifold $\mathcal{P}$) and can be optimized by first order methods.

## C   TRAINING PROCEDURE

In this appendix we discuss key concepts to train the dDGM module. We detail how to back-propagate through the discrete sampling method and introduce an additional loss term to make this possible. We also we provide additional implementation details for updating learnable distance-metric-scaling coefficients and dealing with distance functions during training. Lastly, we describe the two approaches used in this work to make training computationally tractable for larger graphs: symbolic handling of distance metrics and graph subsampling.

### C.1   BACKPROPAGATION THROUGH THE DDGM

The baseline node feature learning part of the architecture is optimized based on the downstream task loss: for classification we use the cross-entropy loss and for regression the mean squared error loss. Nevertheless, we must also update the graph learning dDGM parameters. To do so, we follow the approach proposed by Kazi et al. (2022) and apply a compound loss that rewards edges involved in a correct classification and penalizes edges which result in misclassification. We define the reward function,

$$\delta\left(y_i, \hat{y}_i\right) = \mathbb{E}(ac_i) - ac_i \tag{28}$$

as the difference between the average accuracy of the $i$th sample and the current prediction accuracy, where $y_i$ and $\hat{y}_i$ are the predicted and true labels, and $ac_i = 1$ if $y_i = \hat{y}_i$ or 0 otherwise. Based on $\delta\left(y_i, \hat{y}_i\right)$ we obtain the loss employed to update the graph learning module

$$L_{GL} = \sum_{i=1}^{N}\left(\delta\left(y_i, \hat{y}_i\right)\sum_{l=1}^{l=L}\sum_{j:(i,j)\in\mathcal{E}^{(l)}}\log p_{ij}^{(l)}\right). \tag{29}$$

$L_{GL}$'s gradient approximates the gradient of the expectation

$$\mathbb{E}_{(\mathcal{G}^{(1)},...,\mathcal{G}^{(L)})\sim(\mathbf{P}^{(1)},...,\mathbf{P}^{(L)})}\sum_{i=1}^{N}\delta\left(y_i, \hat{y}_i\right), \tag{30}$$

with respect to the parameters of the graphs in all the layer $\theta_{GL}$. So that,

$$\frac{dL_{GL}}{d\theta_{GL}} \approx \frac{d}{d\theta_{GL}}\mathbb{E}_{(\mathcal{G}^{(1)},...,\mathcal{G}^{(L)})\sim(\mathbf{P}^{(1)},...,\mathbf{P}^{(L)})}\sum_{i=1}^{N}\delta\left(y_i, \hat{y}_i\right). \tag{31}$$

The expectation $\mathbb{E}(ac_i)^{(t)}$ is calculated based on

$$\mathbb{E}(ac_i)^{(t)} = \beta\mathbb{E}(ac_i)^{(t-1)} + (1 - \beta)ac_i, \tag{32}$$

with $\beta = 0.9$ and $\mathbb{E}(ac_i)^{(t=0)} = 0.5$. For regression we use the R2 score instead of the accuracy.

### C.2   UPDATING LEARNABLE DISTANCE-METRIC-SCALING COEFFICIENTS

The scaling metrics are learnable, so that we can indirectly adjust the curvature of each model space without having to backpropagate through the exponential map functions and the formulas for the distances. If we simply take the derivative of the graph loss and update the distance-metric-scaling coefficients during training $\alpha^{(t)} = \alpha^{(t-1)} - lr\frac{\partial L_{GL}}{\partial \alpha^{(t-1)}}$, ($lr$ being the learning rate) this can result in negative values for the coefficients that multiply the distances in different model spaces, which would be mathematically incorrect since distances are by definition positive or zero. To solve this issue we learn $\tilde{\alpha}$, instead of $\alpha$. The two are related by the following equation, $\alpha = S\left(\tilde{\alpha}\right)$, where $S$ is the sigmoid function. So that, $\alpha^{(t)} = S(\tilde{\alpha}^{(t)}) = S\left(\tilde{\alpha}^{(t-1)} - lr\frac{\partial L_{GL}}{\partial \tilde{\alpha}^{(t-1)}}\right)$, which means that

$0 < \alpha^{(t)} < 1$. Using ReLU instead of $S$ could allow $\alpha^{(t)}$ to take arbitrarily large positive values, but we would have gradient problems if $\tilde{\alpha}$ becomes negative. Given that using the sigmoid function bounds the maximum value that the scaling metrics can take, we must multiply the Euclidean plane distance by its own scaling coefficient. Since the Gumbel Top-k trick selects the closest points to generate unweighted edges, rather than storing the actual geodesics, if we scale the Euclidean plane contribution down when necessary, this would equate to having other model spaces with much bigger curvature than that which is actually possible due to the scaling coefficient being now bounded by zero and one. For example, if we were to have a product manifold based on the Cartesian product of the Euclidean plane and a hypersphere, if the model learns a scaling metric close to zero for the Euclidean plane geodesics, this would be equivalent to having a hypersphere with a really large curvature since edges would only be generated based on the distances calculated on the hypersphere. Finally, note that in practice we use the Adam optimizer instead of simple gradient descent for training.

### C.3 COMPUTATIONAL EFFICIENCY

In this section we discuss some of the implementation techniques used to make computation more efficient. We cover two main topics: symbolic handling of distance metrics and graph subsampling. One of the main computational limitations of the approach described in this work is that we must compute distances between all points to generate the latent graph. Although the discrete graph sampling method used by dDGM is more computationally efficient than its continuous counterpart, cDGM — because it generates sparse graphs that make convolutional operators lighter — we quickly run into memory problems for graph datasets with $\mathcal{O}(10^4)$ nodes. Starting from a pointcloud, we must compute the distances between all points to determine whether a connection should be established. This is problematic, since the computational complexity scales quadratically as a function of the number of nodes in the graph, which can rapidly become intractable as we increase the size of our graph.

Most of the experiments were performed using NVIDIA Tesla T4 Tensor Core GPUs with 16 GB of GDDR6 memory, NVIDIA P100 GPUs with 16 GB of CoWoS HBM2 memory, or NVIDIA Tesla K80 GPUs with 24 GB of GDDR5 memory. All these GPUs have limited memories that are easily exceeded during backpropagation for datasets other than Cora and CiteSeer (Yang et al. (2016); Lu & Getoor (2003); Sen et al. (2008)), which have 2,708 nodes and 3,327 nodes, respectively. For example, using the standard PyTorch Geometric implementation we are not able to backpropagate for the PubMed dataset which has 18,717 nodes.

#### C.3.1 SYMBOLIC HANDLING OF DISTANCE METRICS

To avoid memory overflows we resort to Kernel Operations (KeOps) (Charlier et al. (2021)), which makes it possible to compute reductions of large arrays whose entries are given by a mathematical formula. We can classify matrices in three categories: dense, sparse, and symbolic.

Dense matrices are dense numerical arrays $M_{ij} = M[i, j]$ which put a heavy load on computer memory. When they increase in size, they can struggle to fit into RAM or GPU memory. This is what happens in our case, when calculating distances between points. Sparse matrices are typically used to try to address this problem. Sparse matrices use lists of indices $(i_n, j_n)$ and associated values $M_n$. Hence, in sparse matrices we only store the values for non-zero entries. The main limitation of this approach is that the computing speedup obtained by using sparse matrices is highly dependent on sparsity. To obtain significant improvements in performance using sparse encoding, the original matrix should effectively be more than $99\%$ empty (Charlier et al. (2021)). This significantly constrains the applicability of sparse encoding.

Alternatively, KeOps uses symbolic matrices, to represent matrices which can be summarized using an underlying common mathematical structure. In this setup the matrix entries $M_{ij}$ are represented as a function of vectors $\mathbf{x}_i$ and $\mathbf{x}_j$, so that

$$M_{ij} = F(\mathbf{x}_i, \mathbf{x}_j). \tag{33}$$

Even if these objects are not necessarily sparse, they can be represented using only small data arrays $\mathbf{x}_i$ and $\mathbf{x}_j$, which can result into large improvements in computational efficiency, avoiding memory overflow. In our case, the matrices we are working with are fully populated and using sparse tensors

would not enhance performance. On the other hand, the symbolic approach implemented using KeOps enables us to represent the original dense matrix containing all the distances between all the points in a much more compact way, and to work with larger graphs than Cora and Citeseer (Yang et al. (2016); Lu & Getoor (2003); Sen et al. (2008)), namely, PubMed, CS, and Physics (Shchur et al. (2018)). Finally, note that in our case the function used by the symbolic matrix is the distance metric appropiate for whichever manifold we are using to construct the latent graph.

$$M_{ij} = \mathfrak{d}(\mathbf{x}_i, \mathbf{x}_j). \tag{34}$$

### C.3.2 QUANTITATIVE STUDY OF RUNTIME SPEEDUP

We quantify the runtime speedup obtained by using symbolic handling of the distance metrics when generating latent graphs for Cora and CiteSeer. Note, however, that improved execution time is not the only benefit of symbolic handling. As discussed in Section C.3.1, symbolic matrices also help avoid memory overflow for larger graphs. Indeed, this is the reason we choose Cora and CiteSeer to conduct these experiments: using other datasets we experience GPU memory overflow when using dense matrices, and hence we cannot compare dense to symbolic matrix performance. It should also be highlighted that using both dense and symbolic matrices gives the same results in terms of accuracy since they are equivalent mathematically, the difference lies in the computational efficiency of each method.

We display the results in Table 5 and Table 6, in which we record execution times for a 100 epochs using a NVIDIA P100 GPU. We evaluate the effect of increasing the number of dDGM layers on the runtime. We also compare the dDGM module using Euclidean (GCN-dDGM-E) and a product manifold of Euclidean, hyperbolic, and spherical space (GCN-dDGM-EHS) to generate the latent graphs.

Table 5: Results for runtime speedup quantification using symbolic as compared to dense matrices. These results are for the GCN-dDGM-E model using $k = 3$ and training for 100 epochs using NVIDIA P100 GPU.

| Model | No. dDGMs | Matrix | Dataset | Runtime (s) $\pm$ Standard Deviation |
|---|---|---|---|---|
| | 1 | Dense | Cora | $2.98_{\pm 0.01}$ |
| | 1 | Symbolic | Cora | $2.64_{\pm 0.01}$ |
| | 1 | Dense | CiteSeer | $3.50_{\pm 0.02}$ |
| | 1 | Symbolic | CiteSeer | $2.73_{\pm 0.02}$ |
| | 2 | Dense | Cora | $4.87_{\pm 0.19}$ |
| | 2 | Symbolic | Cora | $4.03_{\pm 0.03}$ |
| GCN-dDGM-E | 2 | Dense | CiteSeer | $6.16_{\pm 0.58}$ |
| | 2 | Symbolic | CiteSeer | $4.13_{\pm 0.02}$ |
| | 3 | Dense | Cora | $6.53_{\pm 0.15}$ |
| | 3 | Symbolic | Cora | $5.38_{\pm 0.00}$ |
| | 3 | Dense | CiteSeer | $8.42_{\pm 0.69}$ |
| | 3 | Symbolic | CiteSeer | $5.51_{\pm 0.04}$ |

Table 6: Results for runtime speedup quantification using symbolic as compared to dense matrices. These results are for the GCN-dDGM-EHS model using $k = 3$ and training for 100 epochs using NVIDIA P100 GPU.

| Model | No. dDGMs | Matrix | Dataset | Runtime (s) $\pm$ Standard Deviation |
|---|---|---|---|---|
| | 1 | Dense | Cora | $6.17_{\pm 0.41}$ |
| | 1 | Symbolic | Cora | $4.27_{\pm 0.06}$ |
| | 1 | Dense | CiteSeer | $7.92_{\pm 0.33}$ |
| | 1 | Symbolic | CiteSeer | $4.38_{\pm 0.02}$ |
| | 2 | Dense | Cora | $10.67_{\pm 0.32}$ |
| | 2 | Symbolic | Cora | $7.18_{\pm 0.03}$ |
| GCN-dDGM-EHS | 2 | Dense | CiteSeer | $14.15_{\pm 0.07}$ |
| | 2 | Symbolic | CiteSeer | $7.48_{\pm 0.16}$ |
| | 3 | Dense | Cora | $15.32_{\pm 0.13}$ |
| | 3 | Symbolic | Cora | $10.08_{\pm 0.06}$ |
| | 3 | Dense | CiteSeer | $20.85_{\pm 0.42}$ |
| | 3 | Symbolic | CiteSeer | $10.31_{\pm 0.04}$ |

We can see that as we add more dDGMs the difference between using dense and symbolic matrices becomes more substantial. Likewise, the benefit from using symbolic matrices becomes more apparent when using product manifolds, this is because more distances must be computed. The

product manifolds are calculated based on the Cartesian product of three model spaces and hence, to obtain the overall distance between each of the nodes, we must compute geodesics in all constant curvature manifolds independently. Another observation that we can make is that the computation time for CiteSeer increases substantially as compared to Cora using dense matrices. CiteSeer only has 995 more nodes than Cora, yet using 3 dDGM-EHS layers the execution time for 100 epochs increases by 5.53 seconds for dense matrices. This clearly shows that using dense matrices can quickly become hard to scale for larger graphs, which can have orders of magnitude more nodes than CiteSeer. In line with the literature, using symbolic matrices is more computationally tractable for larger graphs Kazi et al. (2022).

Also, we run some additional experiments to quantify the increase in computation time as a function of $k$, that is, the number of edges per latent graph node when applying the Gumbel Top-k trick. As we can see in Table 7 and Table 8, $k$ does not seem to have a statistically significant impact on the execution time. As before, we find that using symbolic matrices is consistently more efficient.

Table 7: Results for runtime speedup quantification using symbolic as compared to dense matrices for different $k = 1 - 30$. These results are for the GCN-dDGM-E model and training for 100 epochs using a Tesla T4 GPU.

| | | | GCN-dDGM-E |
|---|---|---|---|
| $k$ | Matrix | Dataset | Runtime (s) $\pm$ Standard Deviation |
| 1 | Dense | Cora | $4.22_{\pm 0.14}$ |
| 1 | Symbolic | Cora | $2.84_{\pm 0.02}$ |
| 1 | Dense | CiteSeer | $5.26_{\pm 0.06}$ |
| 1 | Symbolic | CiteSeer | $2.95_{\pm 0.02}$ |
| 2 | Dense | Cora | $4.23_{\pm 0.06}$ |
| 2 | Symbolic | Cora | $2.99_{\pm 0.18}$ |
| 2 | Dense | CiteSeer | $5.29_{\pm 0.03}$ |
| 2 | Symbolic | CiteSeer | $2.98_{\pm 0.01}$ |
| 3 | Dense | Cora | $4.24_{\pm 0.06}$ |
| 3 | Symbolic | Cora | $2.94_{\pm 0.02}$ |
| 3 | Dense | CiteSeer | $5.54_{\pm 0.24}$ |
| 3 | Symbolic | CiteSeer | $3.04_{\pm 0.03}$ |
| 5 | Dense | Cora | $4.25_{\pm 0.11}$ |
| 5 | Symbolic | Cora | $2.93_{\pm 0.01}$ |
| 5 | Dense | CiteSeer | $5.41_{\pm 0.17}$ |
| 5 | Symbolic | CiteSeer | $3.02_{\pm 0.11}$ |
| 7 | Dense | Cora | $4.27_{\pm 0.02}$ |
| 7 | Symbolic | Cora | $2.94_{\pm 0.03}$ |
| 7 | Dense | CiteSeer | $5.44_{\pm 0.16}$ |
| 7 | Symbolic | CiteSeer | $3.11_{\pm 0.18}$ |
| 10 | Dense | Cora | $4.30_{\pm 0.14}$ |
| 10 | Symbolic | Cora | $2.94_{\pm 0.04}$ |
| 10 | Dense | CiteSeer | $5.48_{\pm 0.21}$ |
| 10 | Symbolic | CiteSeer | $3.01_{\pm 0.03}$ |
| 20 | Dense | Cora | $4.67_{\pm 0.52}$ |
| 20 | Symbolic | Cora | $3.19_{\pm 0.05}$ |
| 20 | Dense | CiteSeer | $5.49_{\pm 0.03}$ |
| 20 | Symbolic | CiteSeer | $3.14_{\pm 0.05}$ |
| 30 | Dense | Cora | $4.23_{\pm 0.01}$ |
| 30 | Symbolic | Cora | $3.21_{\pm 0.20}$ |
| 30 | Dense | CiteSeer | $5.25_{\pm 0.03}$ |
| 30 | Symbolic | CiteSeer | $3.35_{\pm 0.03}$ |

Table 8: Results for runtime speedup quantification using symbolic as compared to dense matrices for different $k = 1 - 30$. These results are for the GCN-dDGM-EHS model and training for 100 epochs using a Tesla T4 GPU.

| | | | GCN-dDGM-EHS |
|---|---|---|---|
| $k$ | Matrix | Dataset | Runtime (s) $\pm$ Standard Deviation |
| 1 | Dense | Cora | $5.46_{\pm 0.02}$ |
| 1 | Symbolic | Cora | $4.50_{\pm 0.02}$ |
| 1 | Dense | CiteSeer | $7.27_{\pm 0.80}$ |
| 1 | Symbolic | CiteSeer | $4.97_{\pm 0.56}$ |
| 2 | Dense | Cora | $5.37_{\pm 0.03}$ |
| 2 | Symbolic | Cora | $4.56_{\pm 0.02}$ |
| 2 | Dense | CiteSeer | $6.69_{\pm 0.02}$ |
| 2 | Symbolic | CiteSeer | $4.71_{\pm 0.03}$ |
| 3 | Dense | Cora | $5.36_{\pm 0.02}$ |
| 3 | Symbolic | Cora | $4.57_{\pm 0.16}$ |
| 3 | Dense | CiteSeer | $6.77_{\pm 0.06}$ |
| 3 | Symbolic | CiteSeer | $5.06_{\pm 0.73}$ |
| 5 | Dense | Cora | $5.57_{\pm 0.22}$ |
| 5 | Symbolic | Cora | $4.58_{\pm 0.07}$ |
| 5 | Dense | CiteSeer | $6.74_{\pm 0.02}$ |
| 5 | Symbolic | CiteSeer | $4.70_{\pm 0.03}$ |
| 7 | Dense | Cora | $5.62_{\pm 0.29}$ |
| 7 | Symbolic | Cora | $4.47_{\pm 0.01}$ |
| 7 | Dense | CiteSeer | $6.75_{\pm 0.05}$ |
| 7 | Symbolic | CiteSeer | $4.99_{\pm 0.58}$ |
| 10 | Dense | Cora | $5.92_{\pm 0.73}$ |
| 10 | Symbolic | Cora | $4.70_{\pm 0.04}$ |
| 10 | Dense | CiteSeer | $7.06_{\pm 0.43}$ |
| 10 | Symbolic | CiteSeer | $4.83_{\pm 0.03}$ |
| 20 | Dense | Cora | $5.44_{\pm 0.02}$ |
| 20 | Symbolic | Cora | $4.77_{\pm 0.13}$ |
| 20 | Dense | CiteSeer | $6.76_{\pm 0.03}$ |
| 20 | Symbolic | CiteSeer | $4.86_{\pm 0.18}$ |
| 30 | Dense | Cora | $5.63_{\pm 0.30}$ |
| 30 | Symbolic | Cora | $4.74_{\pm 0.03}$ |
| 30 | Dense | CiteSeer | $6.93_{\pm 0.22}$ |
| 30 | Symbolic | CiteSeer | $5.33_{\pm 0.66}$ |

### C.3.3 TRAINING ON LARGE GRAPHS

Although symbolic handling of distance metrics is certainly necessary, for larger graphs such as the graphs for node property prediction of the Open Graph Benchmark (OGB) (Hu et al. (2020)) which have $\mathcal{O}(10^5) - \mathcal{O}(10^8)$ nodes and $\mathcal{O}(10^6) - \mathcal{O}(10^9)$ edges, we must combine KeOps with graph subsampling techniques to make backpropagating computationally tractable.

We apply a neighbor sampler to track message passing dependencies for the subsampled nodes. This allows computation to be more lightweight. Based on the message passing equation

$$\mathbf{x}_i^{(l+1)} = \phi\Big(\mathbf{x}_i^{(l)}, \bigoplus_{j \in \mathcal{N}(v_i)} \psi(\mathbf{x}_i^{(l)}, \mathbf{x}_j^{(l)})\Big), \tag{35}$$

to calculate $\mathbf{x}_i^{(l+1)}$ we must aggregate, and hence, have stored the node features of its neighbors when subsampling the graph. Note that unlike in the original node prediction setup in which we give as input all the nodes and predict properties for all nodes in the complete graph, here we have a different number of input and output nodes, which gives rise to a bipartite structure for multi-layer minibatch message passing. Such a bipartite graph, which samples only the necessary input and output nodes from the original graph, is called a *message flow graph* (Ladkin & Leue (2005)). For every node that we compute in a given batch we will need to track its message flow graph alongside all relevant dependencies.

## D    THE AEROTHERMODYNAMICS DATASET

We generate a multi-class classification dataset based on Computational Fluid Dynamics (CFD) simulations conducted for the rocket designs developed for the Karman Space Programme. Specifically, the dataset is generated based on the shock wave velocity distribution around the nose of a rocket at 10 degrees of angle of attack. The simulation meshgrid has varying degrees of resolution, and the shock wave is not symmetric due to the angle of attack. Although CFD software uses a meshgrid to discretize space and run the aerothermodynamics simulations, it can be challenging to extract the connectivity of the original graph, since most often the software is designed to only output a point-cloud. Moreover, given that across a shock wave, the static pressure, temperature, and gas density increases almost instantaneously and there is an abrupt decrease in the flow area, the original graph can present high heterophily. Hence, latent graph inference can be beneficial.

For the dataset, we only consider the shock region at the leading edge of the rocket. To obtain the class labels, we separate the shock flow absolute velocity into 4 regions: $\leqslant 300$ m/s, $300 - 450$ m/s, $450 - 650$ m/s and $> 650$ m/s. The original simulation has 207,745 datapoints, but we only focus on the shock around the nose of the rocket and apply graph coarsening, see Figure 4.

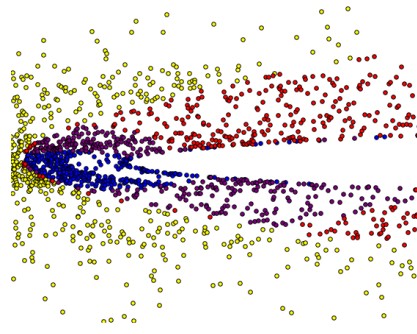

Figure 4: Pointcloud plot of shock intensity regions. The different regions are represented using different colors which correspond to each target class. This is the dataset after applying graph coarsening.

The network input is a pointcloud with pressure values. Note that the coordinates with respect to the rocket are not given to the network. The model is tasked with classifying the shock into four intensity regions which are based on the absolute velocity distribution. Table 9 summarizes the dataset.

Table 9: Summary of properties for Aerothermodynamics dataset

|  | Aerothermodynamics |
| --- | --- |
| Homophily level | N/A |
| Nodes | 1,456 |
| Features | 1 |
| Edges | N/A |
| Classes | 4 |
| Average Degree | N/A |
| Learning | Transductive |
| Task | Classification |
| Network Type | CFD simulation |

# E    ADDITIONAL EXPERIMENTS AND RESULTS

In this appendix we include additional experiments for the latent graph inference system. In Section E.1 we explore using more than one latent graph inference system per GNN model. In Section E.2 we include additional results for the homophilic graph datasets in which we vary the value of $k$ for the graph generation algorithm. In Section E.3 we experiment with using a greater number of model spaces to generate product manifolds for the embedding space. Section E.4 includes results for heterophilic datasets, Section E.5 for OGB, and Section E.6 for inductive learning. The **First**, **Second** and **Third** best models for each dataset are highlighted in each table.

## E.1    NUMBER OF DDGM MODULES

In these experiments, we investigate whether there is any benefit in stacking multiple dDGM modules. Using multiple dDGMs effectively means that within the model, the network layers would be learning based on different latent graphs. In principle, according to the results in Table 10 and 11, there is no clear improvement in performance for Cora and CiteSeer. In fact, the accuracy of the models can decrease. It is only for PubMed that there is sometimes some improvement in the accuracy. These results align with previous studies by Kazi et al. (2022). Finally, we also run a few experiments for Physics and CS. Again, we find no substantial improvement using an additional dDGM module, see Table 12. Given our findings, we use a single dDGM module, since it is more computationally efficient.

Table 10: Results for Cora, CiteSeer, and PubMed using more than one dDGM* (taking a pointcloud as input) latent graph inference module.

|  |  |  | Cora | CiteSeer | PubMed |
| --- | --- | --- | --- | --- | --- |
| Model | $k$ | Layers | Accuracy (%) $\pm$ Standard Deviation | | |
| GCN-dDGM*-E | 3 | dDGM*/GCN/GCN/GCN | $62.48_{+3.24}$ | $62.47_{+3.20}$ | $83.89_{+0.70}$ |
| GCN-dDGM*-E | 3 | dDGM*/GCN/dDGM*/GCN/GCN | $54.22_{+3.79}$ | $59.76_{+2.18}$ | $85.74_{+2.06}$ |
| GCN-dDGM*-EH | 3 | dDGM*/GCN/GCN/GCN | $61.07_{+10.18}$ | $\mathbf{\color{red}65.13_{+2.19}}$ | $86.67_{+0.69}$ |
| GCN-dDGM*-EH | 3 | dDGM*/GCN/dDGM*/GCN/GCN | $51.96_{+4.53}$ | $55.36_{+5.35}$ | $86.19_{+1.10}$ |
| GCN-dDGM*-EHS | 3 | dDGM*/GCN/GCN/GCN | $\mathbf{\color{red}66.87_{+3.27}}$ | $\mathbf{\color{purple}63.80_{+2.94}}$ | $\mathbf{\color{red}87.03_{+0.70}}$ |
| GCN-dDGM*-EHS | 3 | dDGM*/GCN/dDGM*/GCN/GCN | $50.66_{+4.26}$ | $51.66_{+3.24}$ | $86.32_{+0.70}$ |
| GCN-dDGM*-E | 5 | dDGM*/GCN/GCN/GCN | $60.63_{+3.01}$ | $63.28_{+3.35}$ | $86.82_{+0.69}$ |
| GCN-dDGM*-E | 5 | dDGM*/GCN/dDGM*/GCN/GCN | $52.15_{+4.21}$ | $54.46_{+3.19}$ | $79.04_{+4.45}$ |
| GCN-dDGM*-EH | 5 | dDGM*/GCN/GCN/GCN | $\mathbf{\color{purple}63.67_{+6.48}}$ | $61.59_{+12.28}$ | $\mathbf{\color{purple}86.90_{+1.16}}$ |
| GCN-dDGM*-EH | 5 | dDGM*/GCN/dDGM*/GCN/GCN | $45.30_{+3.06}$ | $51.69_{+4.33}$ | $82.02_{+5.42}$ |
| GCN-dDGM*-EHS | 5 | dDGM*/GCN/GCN/GCN | $\mathbf{\color{blue}64.82_{+3.68}}$ | $\mathbf{\color{blue}64.62_{+3.35}}$ | $86.85_{+0.97}$ |
| GCN-dDGM*-EHS | 5 | dDGM*/GCN/dDGM*/GCN/GCN | $45.07_{+4.21}$ | $45.42_{+3.00}$ | $\mathbf{\color{blue}86.92_{+0.83}}$ |
| MLP* | N/A | Linear/Linear/Linear | $58.92_{+3.28}$ | $59.48_{+2.14}$ | $85.75_{+1.02}$ |

Table 11: Results for Cora, CiteSeer, and PubMed using more than one dDGM (leveraging the original graph connectivity structure) latent graph inference module.

| | | | Cora | CiteSeer | PubMed |
|---|---|---|---|---|---|
| Model | $k$ | Layers | \multicolumn{3}{c}{Accuracy (%) $\pm$ Standard Deviation} |
| GCN-dDGM-E | 3 | dDGM/GCN/GCN/GCN | $80.44_{\pm4.60}$ | $70.18_{\pm1.46}$ | $87.45_{\pm0.72}$ |
| GCN-dDGM-E | 3 | dDGM/GCN/dDGM/GCN/GCN | $70.44_{\pm5.81}$ | $69.85_{\pm4.22}$ | $86.67_{\pm0.48}$ |
| GCN-dDGM-EH | 3 | dDGM/GCN/GCN/GCN | $\mathbf{83.65_{\pm5.25}}$ | $\mathbf{72.89_{\pm1.64}}$ | $\mathbf{87.62_{\pm0.64}}$ |
| GCN-dDGM-EH | 3 | dDGM/GCN/dDGM/GCN/GCN | $77.30_{\pm7.30}$ | $72.32_{\pm2.09}$ | $87.02_{\pm0.79}$ |
| GCN-dDGM-EHS | 3 | dDGM/GCN/GCN/GCN | $\mathbf{84.70_{\pm2.96}}$ | $69.98_{\pm2.70}$ | $87.35_{\pm0.90}$ |
| GCN-dDGM-EHS | 3 | dDGM/GCN/dDGM/GCN/GCN | $79.07_{\pm4.35}$ | $71.99_{\pm1.94}$ | $86.93_{\pm0.71}$ |
| GCN-dDGM-E | 5 | dDGM/GCN/GCN/GCN | $82.74_{\pm4.42}$ | $71.78_{\pm1.50}$ | $\mathbf{87.60_{\pm0.69}}$ |
| GCN-dDGM-E | 5 | dDGM/GCN/dDGM/GCN/GCN | $72.70_{\pm6.03}$ | $68.25_{\pm4.57}$ | $86.59_{\pm0.87}$ |
| GCN-dDGM-EH | 5 | dDGM/GCN/GCN/GCN | $82.32_{\pm4.71}$ | $\mathbf{73.56_{\pm2.75}}$ | $\mathbf{87.67_{\pm0.76}}$ |
| GCN-dDGM-EH | 5 | dDGM/GCN/dDGM/GCN/GCN | $76.04_{\pm4.23}$ | $72.11_{\pm2.33}$ | $86.49_{\pm0.73}$ |
| GCN-dDGM-EHS | 5 | dDGM/GCN/GCN/GCN | $\mathbf{83.58_{\pm4.39}}$ | $\mathbf{74.00_{\pm1.68}}$ | $87.12_{\pm0.72}$ |
| GCN-dDGM-EHS | 5 | dDGM/GCN/dDGM/GCN/GCN | $76.67_{\pm5.55}$ | $70.42_{\pm1.66}$ | $82.57_{\pm0.81}$ |
| GCN | N/A | GCN/GCN/GCN | $83.11_{\pm2.29}$ | $69.97_{\pm2.06}$ | $85.75_{\pm1.01}$ |

Table 12: Results for Physics and CS using two dDGM latent graph inference modules with a product manifold combining Euclidean, hyperbolic, and spherical model spaces.

| | | | Physics | CS |
|---|---|---|---|---|
| Model | $k$ | Layers | \multicolumn{2}{c}{Accuracy (%) $\pm$ Standard Deviation} |
| GCN-dDGM-EHS | 5 | dDGM/GCN/GCN/GCN | $\mathbf{96.17_{\pm0.30}}$ | $\mathbf{92.06_{\pm0.83}}$ |
| GCN-dDGM-EHS | 5 | dDGM/GCN/dDGM/GCN/GCN | $\mathbf{96.18_{\pm0.31}}$ | $87.00_{\pm4.00}$ |
| GCN | N/A | GCN/GCN/GCN | $\mathbf{95.51_{\pm0.34}}$ | $\mathbf{87.28_{\pm1.54}}$ |

### E.2 HOMOPHILIC GRAPH DATASETS EXTENDED RESULTS

In this section we include extended results for the Cora, CiteSeer, PubMed, Physics, and CS datasets. Table 13 presents results using single model spaces for the embedding space, and Table 14 and Table 15 using product manifolds. As discussed in the main text, the dDGM* does not use the original dataset graph as inductive bias, since it is only provided with a pointcloud. On the other hand, the dDGM does use the original graph. Different $k$ values are applied.

### E.3 MORE COMPLEX PRODUCT MANIFOLDS RESULTS

The main objective of this section is trying to test the limits of our approach. In Table 16 we display the results multiplying up to five model spaces for the CS dataset. From the results, we can see that adding more product manifolds can result in improved performance. The GCN-dDGM-EHHSS network with $k = 5$ obtains an accuracy of $93.10 \pm 0.74\%$, as compared to the best single model space based model GCN-dDGM-E with $k = 5$ which achieves a result of only $87.88 \pm 2.55\%$.

### E.4 HETEROPHILIC GRAPH DATASETS EXTENDED RESULTS

In Table 17 we display the results using both the dDGM and the dDGM* module for heterophilic datasets. The dDGM module uses the original dataset graph as inductive bias for the generation of the latent graphs. As expected, this leads to worse results than those using the dDGM* for heterophilic graphs, since the original graph is not good for diffusion using GCNs. It is better to start directly from a pointcloud and completely ignore the original adjacency matrix $\mathbf{A}^{(0)}$ since it does not provide the model with a good inductive bias.

### E.5 RESULTS FOR LARGE GRAPHS FROM THE OPEN GRAPH BENCHMARK

Table 18 and Table 19 display results for the OGB-Arxiv and OGB-Products datasets. Although the OGB-Products dataset is considerably larger than the OGB-Arxiv both in terms of the number of nodes and edges, our models achieve slightly better performance. This may be due to the fact that the OGB-Products dataset is an undirected graph, whereas the OGB-Arxiv dataset is directed. The dDGM module generates undirected edges between nodes by construction (if we neglect the noise in the Gumbel Top-k trick), which may be affecting performance in the case of the OGB-Arxiv dataset.

Table 13: Results for classical homophilic datasets combining GCN diffusion layers with the dDGM* and dDGM latent graph inference system and using single model spaces to construct the latent graphs.

| | | Cora | CiteSeer | PubMed | Physics | CS |
|---|---|---|---|---|---|---|
| Homophily level | | 0.81 | 0.74 | 0.80 | 0.93 | 0.80 |
| Nodes | | 2,708 | 3,327 | 18,717 | 34,493 | 18,333 |
| Features | | 1,433 | 3,703 | 500 | 8,415 | 6,805 |
| Edges | | 5,278 | 4,676 | 44,327 | 247,962 | 81,894 |
| Classes | | 7 | 6 | 3 | 5 | 15 |
| Average Degree | | 3.9 | 2.77 | 4.5 | 14.38 | 8.93 |
| Model | $k$ | Accuracy (%) ± Standard Deviation | | | | |
| GCN-dDGM*-E | 3 | **62.48**±3.24 | 62.47±3.20 | 83.89±0.70 | 94.03±0.45 | 76.05±6.89 |
| GCN-dDGM*-H | 3 | **63.82**±3.69 | **63.76**±3.40 | **87.15**±0.69 | 93.24±0.45 | 76.26±6.39 |
| GCN-dDGM*-S | 3 | 32.78±5.90 | 21.96±6.32 | 41.60±5.43 | 58.37±1.44 | 54.52±17.30 |
| GCN-dDGM*-E | 5 | 60.63±3.01 | 63.28±3.35 | **86.82**±0.69 | **95.13**±0.50 | 80.46±2.34 |
| GCN-dDGM*-H | 5 | 61.48±3.68 | **64.76**±2.88 | 85.00±0.69 | 94.83±0.51 | 82.69±1.44 |
| GCN-dDGM*-S | 5 | 42.04±5.01 | 22.20±6.23 | 40.38±5.44 | 54.70±2.01 | 70.67±13.69 |
| GCN-dDGM*-E | 7 | **62.30**±5.27 | 61.36±7.46 | **86.86**±0.68 | 93.71±3.23 | **86.03**±5.27 |
| GCN-dDGM*-H | 7 | 61.44±10.83 | 62.05±5.74 | 85.19±5.07 | **95.05**±0.30 | **84.20**±4.70 |
| GCN-dDGM*-S | 7 | 41.71±14.22 | 20.75±2.84 | 40.49±1.57 | 55.20±13.26 | 73.46±8.17 |
| GCN-dDGM*-E | 10 | 61.37±3.85 | **64.27**±3.97 | 85.48±4.40 | 91.82±3.62 | 81.42±4.60 |
| GCN-dDGM*-H | 10 | 61.63±4.56 | 63.62±5.22 | 84.92±4.85 | **95.02**±0.46 | 80.36±5.42 |
| GCN-dDGM*-S | 10 | 34.04±9.65 | 19.96±4.14 | 39.84±1.07 | 50.52±2.77 | 69.55±5.94 |
| MLP* | N/A | 58.92±3.28 | 59.48±2.14 | 85.75±1.02 | 94.91±0.30 | **87.80**±1.54 |
| | | Cora | CiteSeer | PubMed | Physics | CS |
| GCN-dDGM-E | 3 | 80.44±4.60 | 70.18±1.46 | 87.45±0.72 | **96.03**±0.41 | 85.98±2.59 |
| GCN-dDGM-H | 3 | 80.31±4.30 | 69.78±1.56 | 85.79±0.73 | 95.29±0.43 | 85.95±2.58 |
| GCN-dDGM-S | 3 | 73.26±4.33 | 67.98±2.21 | 81.39±1.00 | 91.60±0.55 | 76.76±4.65 |
| GCN-dDGM-E | 5 | 82.74±4.42 | 71.78±1.50 | **87.60**±0.69 | 95.96±0.40 | 87.88±2.55 |
| GCN-dDGM-H | 5 | 80.80±4.43 | 67.53±2.12 | 87.58±0.71 | **96.06**±0.42 | 87.65±2.56 |
| GCN-dDGM-S | 5 | 80.81±2.30 | 71.81±1.20 | 77.65±1.01 | 95.91±0.41 | 80.73±4.62 |
| GCN-dDGM-E | 7 | 82.11±4.24 | **72.35**±1.92 | **87.69**±0.67 | 95.50±1.25 | 87.17±3.82 |
| GCN-dDGM-H | 7 | **84.68**±3.31 | 70.43±4.95 | **87.74**±0.72 | **96.06**±0.46 | **88.78**±2.24 |
| GCN-dDGM-S | 7 | 80.44±5.26 | **72.89**±2.00 | 87.13±0.66 | 95.76±0.43 | 84.16±2.78 |
| GCN-dDGM-E | 10 | 81.67±5.74 | **72.44**±2.81 | 85.45±4.32 | 95.99±0.49 | 88.19±3.51 |
| GCN-dDGM-H | 10 | **83.15**±3.20 | 71.54±1.46 | **87.69**±0.76 | 95.84±0.73 | **88.49**±2.21 |
| GCN-dDGM-S | 10 | 82.70±3.22 | 72.29±2.87 | 87.38±0.69 | **96.03**±0.44 | 85.20±4.07 |
| GCN | N/A | **83.11**±2.29 | 69.97±2.06 | 85.75±1.01 | 95.51±0.34 | 87.28±1.54 |

Table 14: Results for classical homophilic datasets combining GCN diffusion layers with the dDGM* module and using product manifolds to construct the latent graphs.

| | | Cora | CiteSeer | PubMed | Physics | CS |
|---|---|---|---|---|---|---|
| Homophily level | | 0.81 | 0.74 | 0.80 | 0.93 | 0.80 |
| Nodes | | 2,708 | 3,327 | 18,717 | 34,493 | 18,333 |
| Features | | 1,433 | 3,703 | 500 | 8,415 | 6,805 |
| Edges | | 5,278 | 4,676 | 44,327 | 247,962 | 81,894 |
| Classes | | 7 | 6 | 3 | 5 | 15 |
| Average Degree | | 3.9 | 2.77 | 4.5 | 14.38 | 8.93 |
| Model | $k$ | Accuracy (%) ± Standard Deviation | | | | |
| GCN-dDGM*-HH | 3 | 57.00±9.78 | 62.50±4.25 | **87.43**±0.40 | 92.20±3.47 | 74.88±4.65 |
| GCN-dDGM*-SS | 3 | 38.26±10.94 | 24.20±10.12 | 41.59±0.71 | 54.43±10.00 | 55.29±7.33 |
| GCN-dDGM*-EH | 3 | 61.07±10.18 | 65.13±2.19 | 86.67±0.69 | 94.85±0.53 | 85.91±2.88 |
| GCN-dDGM*-ES | 3 | 62.07±4.08 | 64.31±3.15 | 86.85±1.02 | 95.01±0.55 | 78.67±4.44 |
| GCN-dDGM*-HS | 3 | 59.55±10.90 | 63.95±2.35 | **87.04**±0.79 | 95.02±0.45 | 76.55±9.89 |
| GCN-dDGM*-EHH | 3 | **70.85**±4.30 | **68.86**±2.97 | 39.93±1.35 | **95.21**±0.34 | **92.22**±1.09 |
| GCN-dDGM*-EHS | 3 | 66.87±3.27 | 63.80±2.94 | **87.03**±0.70 | 95.03±0.39 | 88.64±1.90 |
| GCN-dDGM*-HH | 5 | 59.03±4.86 | 63.47±1.70 | 86.30±0.99 | 94.31±0.47 | 76.48±4.30 |
| GCN-dDGM*-SS | 5 | 41.33±10.54 | 21.48±3.81 | 40.70±1.31 | 53.43±9.15 | 58.47±5.79 |
| GCN-dDGM*-EH | 5 | 63.67±6.48 | 61.59±12.28 | 86.90±1.16 | 93.94±3.37 | 86.00±3.04 |
| GCN-dDGM*-ES | 5 | 63.44±4.18 | 63.10±2.71 | 86.83±0.82 | 93.92±2.38 | 75.86±11.18 |
| GCN-dDGM*-HS | 5 | 64.44±4.00 | 62.80±3.90 | 85.17±4.72 | 94.81±0.23 | 77.93±4.47 |
| GCN-dDGM*-EHH | 5 | **69.74**±5.09 | **66.81**±2.04 | 39.93±1.35 | **95.25**±0.36 | **90.46**±1.03 |
| GCN-dDGM*-EHS | 5 | 64.82±3.68 | 64.62±3.35 | 86.85±0.97 | 94.30±2.09 | 88.48±1.77 |
| GCN-dDGM*-HH | 7 | 54.58±10.40 | 62.02±5.73 | 86.91±0.89 | 94.08±3.14 | 79.88±5.77 |
| GCN-dDGM*-SS | 7 | 40.22±13.00 | 20.54±3.42 | 40.49±1.39 | 58.67±16.44 | 50.94±14.36 |
| GCN-dDGM*-EH | 7 | 59.92±8.92 | 64.79±2.32 | 86.97±0.45 | 94.18±2.46 | 80.82±6.38 |
| GCN-dDGM*-ES | 7 | 61.44±3.23 | 62.41±2.88 | 86.48±0.58 | 92.13±5.81 | 74.84±7.24 |
| GCN-dDGM*-HS | 7 | 59.11±8.15 | 64.86±3.03 | 85.21±4.74 | 94.72±0.92 | 83.23±3.95 |
| GCN-dDGM*-EHH | 7 | **70.37**±4.72 | 62.50±11.69 | 39.93±1.35 | **95.16**±0.37 | **91.58**±0.91 |
| GCN-dDGM*-EHS | 7 | 63.30±3.93 | 64.54±1.89 | 86.99±0.79 | 92.85±2.49 | 86.96±2.54 |
| GCN-dDGM*-HH | 10 | 57.59±8.05 | 59.43±5.46 | 86.43±0.70 | 92.50±3.35 | 73.13±8.55 |
| GCN-dDGM*-SS | 10 | 36.30±8.23 | 21.20±2.88 | 39.83±1.17 | 50.52±2.77 | 53.85±16.78 |
| GCN-dDGM*-EH | 10 | 59.63±11.04 | **65.49**±2.86 | 84.54±5.77 | 93.55±4.53 | 72.79±5.79 |
| GCN-dDGM*-ES | 10 | 63.04±4.21 | 62.65±6.29 | 86.71±0.89 | 93.42±3.22 | 74.85±7.59 |
| GCN-dDGM*-HS | 10 | 64.24±5.38 | 62.73±2.59 | 85.24±4.58 | 95.03±0.64 | 75.03±6.18 |
| GCN-dDGM*-EHH | 10 | 69.63±4.00 | 64.70±4.61 | 39.93±1.35 | 95.02±0.39 | 83.81±11.41 |
| GCN-dDGM*-EHS | 10 | 63.61±3.71 | 64.52±2.77 | 86.60±0.66 | 90.03±4.86 | 84.52±7.39 |
| MLP* | N/A | 58.92±3.28 | 59.48±2.14 | 85.75±1.02 | 94.91±0.30 | 87.80±1.54 |

Table 15: Results for classical homophilic datasets combining GCN diffusion layers with the dDGM module and using product manifold to construct the latent graphs.

|  | | **Cora** | **CiteSeer** | **PubMed** | **Physics** | **CS** |
|---|---|---|---|---|---|---|
| Homophily level | | 0.81 | 0.74 | 0.80 | 0.93 | 0.80 |
| Nodes | | 2,708 | 3,327 | 18,717 | 34,493 | 18,333 |
| Features | | 1,433 | 3,703 | 500 | 8,415 | 6,805 |
| Edges | | 5,278 | 4,676 | 44,327 | 247,962 | 81,894 |
| Classes | | 7 | 6 | 3 | 5 | 15 |
| Average Degree | | 3.9 | 2.77 | 4.5 | 14.38 | 8.93 |
| Model | $k$ | | | Accuracy (%) $\pm$ Standard Deviation | | |
| GCN-dDGM-HH | 3 | $77.82_{\pm 5.46}$ | $71.27_{\pm 2.09}$ | $87.35_{\pm 0.65}$ | $94.04_{\pm 3.54}$ | $82.91_{\pm 3.00}$ |
| GCN-dDGM-SS | 3 | $67.00_{\pm 10.23}$ | $59.16_{\pm 5.96}$ | $86.83_{\pm 0.60}$ | $90.30_{\pm 5.90}$ | $59.31_{\pm 7.18}$ |
| GCN-dDGM-EH | 3 | $83.65_{\pm 5.25}$ | $72.89_{\pm 1.64}$ | $87.62_{\pm 0.64}$ | $96.07_{\pm 0.27}$ | $91.37_{\pm 1.28}$ |
| GCN-dDGM-ES | 3 | $81.19_{\pm 6.63}$ | $71.87_{\pm 3.20}$ | $86.47_{\pm 2.30}$ | $95.32_{\pm 0.42}$ | $90.87_{\pm 0.82}$ |
| GCN-dDGM-HS | 3 | $80.70_{\pm 2.96}$ | $72.77_{\pm 2.76}$ | $87.29_{\pm 0.85}$ | $96.10_{\pm 0.35}$ | $89.43_{\pm 2.37}$ |
| GCN-dDGM-EHH | 3 | $\mathbf{86.59_{\pm 3.33}}$ | $\mathbf{75.42_{\pm 2.39}}$ | $49.17_{\pm 19.39}$ | $96.06_{\pm 0.34}$ | $\mathbf{92.86_{\pm 0.96}}$ |
| GCN-dDGM-EHS | 3 | $\mathbf{85.84_{\pm 2.96}}$ | $69.98_{\pm 2.70}$ | $87.35_{\pm 0.90}$ | $95.96_{\pm 0.42}$ | $89.93_{\pm 3.86}$ |
| GCN-dDGM-HH | 5 | $76.09_{\pm 7.11}$ | $72.65_{\pm 2.23}$ | $87.36_{\pm 0.78}$ | $95.93_{\pm 0.37}$ | $80.73_{\pm 4.62}$ |
| GCN-dDGM-SS | 5 | $65.96_{\pm 9.46}$ | $65.37_{\pm 5.90}$ | $87.21_{\pm 0.75}$ | $90.40_{\pm 6.90}$ | $70.60_{\pm 9.31}$ |
| GCN-dDGM-EH | 5 | $82.32_{\pm 4.71}$ | $73.56_{\pm 2.75}$ | $\mathbf{87.67_{\pm 0.76}}$ | $94.48_{\pm 4.33}$ | $91.11_{\pm 1.57}$ |
| GCN-dDGM-ES | 5 | $81.44_{\pm 5.80}$ | $71.57_{\pm 2.08}$ | $87.41_{\pm 1.14}$ | $96.11_{\pm 0.40}$ | $89.31_{\pm 2.31}$ |
| GCN-dDGM-HS | 5 | $82.59_{\pm 4.50}$ | $72.51_{\pm 2.52}$ | $87.79_{\pm 1.08}$ | $95.13_{\pm 2.41}$ | $90.98_{\pm 1.00}$ |
| GCN-dDGM-EHH | 5 | $\mathbf{86.63_{\pm 3.25}}$ | $73.95_{\pm 2.97}$ | $44.53_{\pm 13.84}$ | $96.16_{\pm 0.37}$ | $\mathbf{92.30_{\pm 1.05}}$ |
| GCN-dDGM-EHS | 5 | $83.58_{\pm 4.39}$ | $74.00_{\pm 1.68}$ | $87.12_{\pm 0.72}$ | $\mathbf{96.17_{\pm 0.30}}$ | $\mathbf{92.06_{\pm 0.83}}$ |
| GCN-dDGM-HH | 7 | $79.70_{\pm 5.55}$ | $71.30_{\pm 2.94}$ | $85.08_{\pm 5.15}$ | $95.99_{\pm 0.44}$ | $86.27_{\pm 2.66}$ |
| GCN-dDGM-SS | 7 | $63.62_{\pm 9.19}$ | $61.39_{\pm 5.61}$ | $87.41_{\pm 0.55}$ | $91.68_{\pm 5.50}$ | $64.05_{\pm 13.45}$ |
| GCN-dDGM-EH | 7 | $82.89_{\pm 4.31}$ | $71.84_{\pm 2.50}$ | $87.40_{\pm 0.59}$ | $96.06_{\pm 0.52}$ | $91.22_{\pm 0.93}$ |
| GCN-dDGM-ES | 7 | $80.85_{\pm 4.75}$ | $71.08_{\pm 4.14}$ | $85.54_{\pm 5.04}$ | $95.24_{\pm 2.88}$ | $89.21_{\pm 2.56}$ |
| GCN-dDGM-HS | 7 | $81.78_{\pm 5.25}$ | $72.41_{\pm 1.86}$ | $87.27_{\pm 0.62}$ | $96.03_{\pm 0.37}$ | $89.90_{\pm 1.06}$ |
| GCN-dDGM-EHH | 7 | $85.65_{\pm 3.54}$ | $\mathbf{74.67_{\pm 2.01}}$ | $48.04_{\pm 16.30}$ | $\mathbf{96.18_{\pm 0.39}}$ | $90.74_{\pm 2.88}$ |
| GCN-dDGM-EHS | 7 | $83.70_{\pm 3.88}$ | $73.02_{\pm 2.57}$ | $87.61_{\pm 0.67}$ | $96.09_{\pm 0.38}$ | $90.64_{\pm 1.26}$ |
| GCN-dDGM-HH | 10 | $82.18_{\pm 2.90}$ | $72.47_{\pm 2.81}$ | $87.50_{\pm 0.91}$ | $94.73_{\pm 2.83}$ | $86.48_{\pm 0.88}$ |
| GCN-dDGM-SS | 10 | $66.15_{\pm 1.61}$ | $61.51_{\pm 8.14}$ | $\mathbf{87.82_{\pm 0.59}}$ | $90.72_{\pm 5.26}$ | $69.62_{\pm 10.14}$ |
| GCN-dDGM-EH | 10 | $81.44_{\pm 3.94}$ | $73.10_{\pm 2.26}$ | $87.41_{\pm 0.80}$ | $96.03_{\pm 0.37}$ | $90.98_{\pm 1.15}$ |
| GCN-dDGM-ES | 10 | $80.07_{\pm 5.40}$ | $72.86_{\pm 2.51}$ | $87.50_{\pm 0.65}$ | $95.41_{\pm 1.73}$ | $90.25_{\pm 1.79}$ |
| GCN-dDGM-HS | 10 | $83.78_{\pm 3.32}$ | $72.52_{\pm 2.82}$ | $85.89_{\pm 4.29}$ | $95.84_{\pm 0.29}$ | $88.64_{\pm 2.97}$ |
| GCN-dDGM-EHH | 10 | $84.53_{\pm 3.84}$ | $\mathbf{74.33_{\pm 2.30}}$ | $39.93_{\pm 1.35}$ | $95.63_{\pm 1.36}$ | $91.54_{\pm 2.09}$ |
| GCN-dDGM-EHS | 10 | $83.50_{\pm 4.64}$ | $71.82_{\pm 1.68}$ | $87.05_{\pm 1.38}$ | $\mathbf{96.21_{\pm 0.44}}$ | $91.42_{\pm 1.08}$ |
| GCN | N/A | $83.11_{\pm 2.29}$ | $69.97_{\pm 2.06}$ | $85.75_{\pm 1.01}$ | $95.51_{\pm 0.34}$ | $87.28_{\pm 1.54}$ |

Table 16: Results using more complex product manifolds for the CS dataset. We multiply up to five model spaces to generate the product manifolds.

|  | | **CS** |
|---|---|---|
| Model | $k$ | Accuracy (%) $\pm$ Standard Deviation |
| GCN-dDGM-E | 3 | $85.98_{\pm 2.59}$ |
| GCN-dDGM-H | 3 | $85.95_{\pm 2.58}$ |
| GCN-dDGM-S | 3 | $76.76_{\pm 4.65}$ |
| GCN-dDGM-EH | 3 | $91.37_{\pm 1.28}$ |
| GCN-dDGM-ES | 3 | $90.87_{\pm 0.82}$ |
| GCN-dDGM-HS | 3 | $89.43_{\pm 2.37}$ |
| GCN-dDGM-EHH | 3 | $92.86_{\pm 0.96}$ |
| GCN-dDGM-EHS | 3 | $89.93_{\pm 3.86}$ |
| GCN-dDGM-EHHH | 3 | $92.86_{\pm 1.04}$ |
| GCN-dDGM-EHHS | 3 | $92.70_{\pm 0.66}$ |
| GCN-dDGM-EHHSH | 3 | $\mathbf{92.96_{\pm 0.46}}$ |
| GCN-dDGM-EHHSS | 3 | $91.51_{\pm 2.51}$ |
| GCN-dDGM-E | 5 | $87.88_{\pm 2.55}$ |
| GCN-dDGM-H | 5 | $87.65_{\pm 2.56}$ |
| GCN-dDGM-S | 5 | $80.73_{\pm 4.62}$ |
| GCN-dDGM-EH | 5 | $91.11_{\pm 1.57}$ |
| GCN-dDGM-ES | 5 | $89.31_{\pm 2.31}$ |
| GCN-dDGM-HS | 5 | $90.98_{\pm 1.00}$ |
| GCN-dDGM-EHH | 5 | $92.30_{\pm 1.05}$ |
| GCN-dDGM-EHS | 5 | $92.06_{\pm 0.83}$ |
| GCN-dDGM-EHHH | 5 | $92.63_{\pm 0.63}$ |
| GCN-dDGM-EHHS | 5 | $92.82_{\pm 1.34}$ |
| GCN-dDGM-EHHSH | 5 | $\mathbf{92.91_{\pm 0.66}}$ |
| GCN-dDGM-EHHSS | 5 | $\mathbf{93.10_{\pm 0.74}}$ |
| GCN | N/A | $87.28_{\pm 1.54}$ |

Table 17: Results for heterophilic datasets combining GCN diffusion layers with the dDGM* latent graph inference system. We display results using model spaces as well as product manifolds to construct the latent graphs.

| | Texas | Wisconsin | Squirrel | Chameleon |
|---|---|---|---|---|
| Homophily level | 0.11 | 0.21 | 0.22 | 0.23 |
| Nodes | 183 | 251 | 5,201 | 2,277 |
| Features | 1,703 | 1,703 | 2,089 | 2,325 |
| Edges | 295 | 466 | 198,498 | 31,421 |
| Classes | 5 | 5 | 5 | 5 |
| Average Degree | 3.22 | 3.71 | 76.33 | 27.60 |
| $k$ | 2 | 10 | 3 | 5 |
| Model | Accuracy (%) $\pm$ Standard Deviation | | | |
| GCN-dDGM*-E | $\mathbf{80.00}_{+8.31}$ | $\mathbf{88.00}_{+5.65}$ | $34.35_{+2.34}$ | $\mathbf{48.90}_{+3.61}$ |
| GCN-dDGM*-H | $79.44_{+7.88}$ | $\mathbf{89.03}_{+1.89}$ | $\mathbf{35.00}_{+2.35}$ | $48.28_{+4.11}$ |
| GCN-dDGM*-S | $73.88_{+9.95}$ | $85.33_{+4.98}$ | $33.12_{+2.22}$ | $\mathbf{48.63}_{+3.12}$ |
| GCN-dDGM*-HH | $78.89_{+8.53}$ | $\mathbf{88.00}_{+3.26}$ | $\mathbf{34.38}_{+1.07}$ | $\mathbf{48.33}_{+4.14}$ |
| GCN-dDGM*-SS | $73.89_{+8.62}$ | $74.66_{+18.85}$ | $34.06_{+2.20}$ | $48.28_{+3.07}$ |
| GCN-dDGM*-EH | $\mathbf{81.67}_{+7.05}$ | $86.67_{+3.77}$ | $34.37_{+1.72}$ | $47.58_{+3.85}$ |
| GCN-dDGM*-ES | $\mathbf{81.11}_{+10.30}$ | $76.00_{+11.31}$ | $33.38_{+1.86}$ | $47.49_{+3.60}$ |
| GCN-dDGM*-HS | $\mathbf{81.11}_{+9.69}$ | $86.67_{+1.89}$ | $\mathbf{34.65}_{+2.45}$ | $47.84_{+2.67}$ |
| GCN-dDGM*-EHH | $\mathbf{81.11}_{+5.09}$ | $77.60_{+8.62}$ | $33.19_{+1.92}$ | $44.27_{+2.96}$ |
| GCN-dDGM*-EHS | $79.44_{+6.11}$ | $\mathbf{89.33}_{+1.89}$ | $34.17_{+2.23}$ | $47.58_{+4.54}$ |
| GCN-dDGM-E | $60.56_{+8.03}$ | $70.67_{+10.49}$ | $29.87_{+2.46}$ | $44.19_{+3.85}$ |
| GCN-dDGM-H | $58.89_{+9.36}$ | $72.00_{+5.56}$ | $29.56_{+2.49}$ | $44.01_{+4.08}$ |
| GCN-dDGM-S | $59.44_{+8.62}$ | $62.66_{+16.11}$ | $30.58_{+2.34}$ | $45.46_{+2.35}$ |
| GCN-dDGM-HH | $57.22_{+5.58}$ | $60.00_{+19.87}$ | $30.29_{+1.37}$ | $44.19_{+3.78}$ |
| GCN-dDGM-SS | $59.44_{+6.11}$ | $49.33_{+12.36}$ | $30.15_{+2.40}$ | $45.29_{+1.87}$ |
| GCN-dDGM-EH | $62.78_{+9.31}$ | $65.33_{+4.99}$ | $30.00_{+2.58}$ | $43.09_{+3.42}$ |
| GCN-dDGM-ES | $60.56_{+8.03}$ | $69.33_{+6.80}$ | $30.44_{+2.38}$ | $45.68_{+2.66}$ |
| GCN-dDGM-HS | $57.78_{+10.00}$ | $72.00_{+3.26}$ | $30.06_{+2.66}$ | $43.30_{+4.67}$ |
| GCN-dDGM-EHH | $57.77_{+10.88}$ | $44.80_{+10.55}$ | $28.55_{+4.28}$ | $41.01_{+7.68}$ |
| GCN-dDGM-EHS | $58.89_{+7.53}$ | $76.00_{+3.26}$ | $30.27_{+2.95}$ | $41.15_{+9.84}$ |
| MLP* | $77.78_{+10.24}$ | $85.33_{+4.99}$ | $30.44_{+2.55}$ | $40.35_{+3.37}$ |
| GCN | $41.66_{+11.72}$ | $47.20_{+9.76}$ | $24.19_{+2.56}$ | $32.56_{+3.53}$ |

Table 18: Results for OGB-Arxiv dataset using GATv2 diffusion layers and different latent graph inference modules.

| | OGB-Arxiv |
|---|---|
| Nodes | 169,343 |
| Features | 128 |
| Edges | 1,166,243 |
| Classes | 40 |
| $k$ | 20 |
| Model | Accuracy (%) $\pm$ Standard Deviation |
| GATv2-dDGM*-E | $64.34_{+0.40}$ |
| GATv2-dDGM*-H | $61.30_{+0.50}$ |
| GATv2-dDGM*-S | $64.41_{+0.64}$ |
| GATv2-dDGM*-HH | $64.24_{+0.17}$ |
| GATv2-dDGM*-SS | $64.05_{+0.29}$ |
| GATv2-dDGM*-EH | $64.08_{+1.01}$ |
| GATv2-dDGM*-ES | $64.13_{+0.61}$ |
| GATv2-dDGM*-HS | $64.45_{+0.12}$ |
| GATv2-dDGM*-EHS | $\mathbf{65.06}_{+0.09}$ |
| GATv2-dDGM-E | $\mathbf{64.65}_{+0.01}$ |
| GATv2-dDGM-H | $\mathbf{65.05}_{+0.10}$ |
| GATv2-dDGM-S | $64.60_{+0.16}$ |
| GATv2-dDGM-HH | $64.00_{+0.36}$ |
| GATv2-dDGM-SS | $64.35_{+0.36}$ |
| GATv2-dDGM-EH | $64.00_{+0.76}$ |
| GATv2-dDGM-ES | $64.37_{+0.04}$ |
| GATv2-dDGM-HS | $61.00_{+1.12}$ |
| GATv2-dDGM-EHS | $64.25_{+0.50}$ |
| MLP* | $63.49_{+0.15}$ |
| GATv2 | $61.93_{+1.62}$ |

Table 19: Results for OGB-Products dataset using GATv2 diffusion layers and different latent graph inference modules.

| | | **OGB-Products** |
|---|---|---|
| Nodes | | 2,449,029 |
| Features | | 100 |
| Edges | | 61,859,140 |
| Classes | | 47 |
| Model | $k$ | Accuracy (%) ± Standard Deviation |
| GATv2-dDGM-E | 3 | **66.59**±**0.30** |
| GATv2-dDGM-H | 3 | 62.22±0.25 |
| GATv2-dDGM-EH | 3 | 65.51±0.30 |
| GATv2-dDGM-E | 5 | **66.25**±**0.71** |
| GATv2-dDGM-H | 5 | 63.95±0.42 |
| GATv2-dDGM-EH | 5 | 65.62±0.20 |
| MLP* | N/A | **66.05**±**0.20** |
| GATv2 | N/A | 62.02±2.60 |

### E.6 INDUCTIVE LEARNING: RESULTS FOR THE QM9 AND ALCHEMY DATASETS

The datasets discussed in the main task were solely concerned with tranductive learning. For completeness, we show that the latent graph inference system based on product manifolds is also applicable to inductive learning. Molecules are naturally represented as graphs, with atoms as nodes and bonds as edges. Prediction of molecular properties is a popular application of GNNs in chemistry Wieder et al. (2020); Stark et al. (2021); Li et al. (2021); Godwin et al. (2021); Zhang et al. (2021). Specifically, we work with the QM9 (Ramakrishnan et al. (2014); Ruddigkeit et al. (2012)) and Alchemy (Morris et al. (2020)) datasets which are well known in the Geometric Deep Learning literature. Table 20 displays the results. These tasks are substantially different to the ones previously discussed because they involve inductive learning and regression, whereas before all tasks focused on transductive learning and multi-class classification.

Table 20: Results for the QM9 and Alchemy datasets using the dDGM module.

| | **QM9** | **Alchemy** |
|---|---|---|
| No. graphs | 133,885 | 119,487 |
| Targets | 12 | 12 |
| $k$ | 5 | 5 |
| Model | R2 score (×100) ± Standard Deviation | |
| GCN-dDGM*-E | 96.23±1.55 | 96.11±1.43 |
| GCN-dDGM*-H | 97.50±1.04 | **96.12**±**1.59** |
| GCN-dDGM*-S | 96.52±2.30 | 94.52±2.45 |
| GCN-dDGM*-HH | 96.53±2.03 | 95.06±1.59 |
| GCN-dDGM*-SS | 95.51±1.23 | 92.01±2.04 |
| GCN-dDGM*-EH | **97.79**±**1.24** | **96.52**±**1.89** |
| GCN-dDGM*-ES | 94.03±2.13 | 94.10±2.54 |
| GCN-dDGM*-HS | 96.59±1.53 | 96.00±1.01 |
| GCN-dDGM*-EHS | 96.78±1.56 | 96.03±2.59 |
| GCN-dDGM-E | **97.79**±**1.34** | 96.10±2.01 |
| GCN-dDGM-H | 96.69±1.34 | 96.11±1.04 |
| GCN-dDGM-S | 95.53±2.30 | 91.98±2.40 |
| GCN-dDGM-HH | 96.78±1.05 | **96.45**±**2.54** |
| GCN-dDGM-SS | 96.45±2.32 | 93.67±3.01 |
| GCN-dDGM-EH | **98.00**±**1.34** | 96.04±1.45 |
| GCN-dDGM-ES | 96.52±1.54 | 95.02±1.23 |
| GCN-dDGM-HS | 96.51±1.23 | 95.39±1.01 |
| GCN-dDGM-EHS | **97.99**±**1.02** | 96.02±2.10 |
| MLP* | 82.20±3.80 | 76.56±1.50 |
| GCN | 94.35±1.30 | 95.15±1.43 |

## F  LATENT GRAPH LEARNING PLOTS

In this appendix we include additional plots for the learned latent graphs for the heterophilic datasets discussed in the main text as well as for the TadPole and the Aerothermodynamics dataset.

### F.1 LEARNED LATENT GRAPHS FOR HETEROPHILIC DATASETS

In Figure 5, 6, 7, and 8, we display the original graphs provided by the heterophilic datasets, and compare them to the latent graphs generated by the dDGM modules. From the plots we can clearly see the high homophily levels of the Texas latent graph in Figure 5, for which we obtain four distinct clusters. In the case of the bigger datasets in Figure 7, and 8, the algorithm is still able to create clusters but there is mixing between classes.

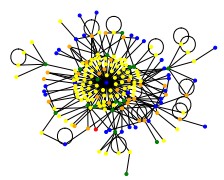 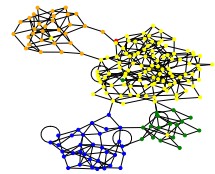

(a) Original graph, $h = 0.11$.        (b) Learned latent graph, $h = 0.93$.

Figure 5: Texas, original vs learned latent graph. The learned latent graph displayed was learned using the GCN-dDGM*-EH model with $k = 2$ in the Gumbel Top-k trick.

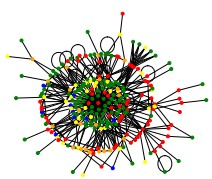

(a) Original graph, $h = 0.21$.        (b) Learned latent graph, $h = 0.64$.

Figure 6: Wisconsin, original vs learned latent graph. The learned latent graph displayed was learned using the GCN-dDGM*-EHS model with $k = 10$ in the Gumbel Top-k trick.

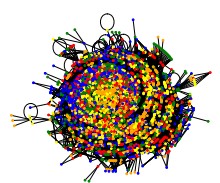

(a) Original graph, $h = 0.22$.        (b) Learned latent graph, $h = 0.32$.

Figure 7: Squirrel, original vs learned latent graph. The learned latent graph displayed was learned using the GCN-dDGM*-S model with $k = 3$ in the Gumbel Top-k trick. Note that both graphs (a) and (b) have the same number of nodes, but in (a) nodes are displayed more closely packed due to the connectivity structure of the graph.

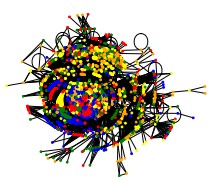 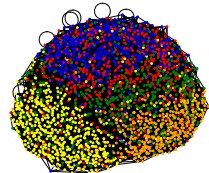

(a) Original graph, $h = 0.23$.        (b) Learned latent graph, $h = 0.42$.

Figure 8: Chameleon, original vs learned latent graph. The learned latent graph displayed was learned using the GCN-dDGM*-E model with $k = 3$ in the Gumbel Top-k trick. Note that both graphs (a) and (b) have the same number of nodes, but in (a) nodes are displayed more closely packed due to the connectivity structure of the graph.

Note that the generated latent graphs are dependent on both the downstream task and the diffusion layers we are using, that is, the GCNs. Since we have created a fully-differentiable system the models are optimizing the latent graph generation together with the rest of the model parameters. Hence, if we were to change the downstream task or the diffusion layers we would expect different latent graphs.

In Figure 2 from Section 4.1 we showed the latent graph learning evolution plots as a function of training epochs for the Texas dataset. We provide additional plots for other datasets. This shows that the latent graph inference system is applicable across a wide range of datasets and that is learns to organize the connectivity of the latent graph during training.

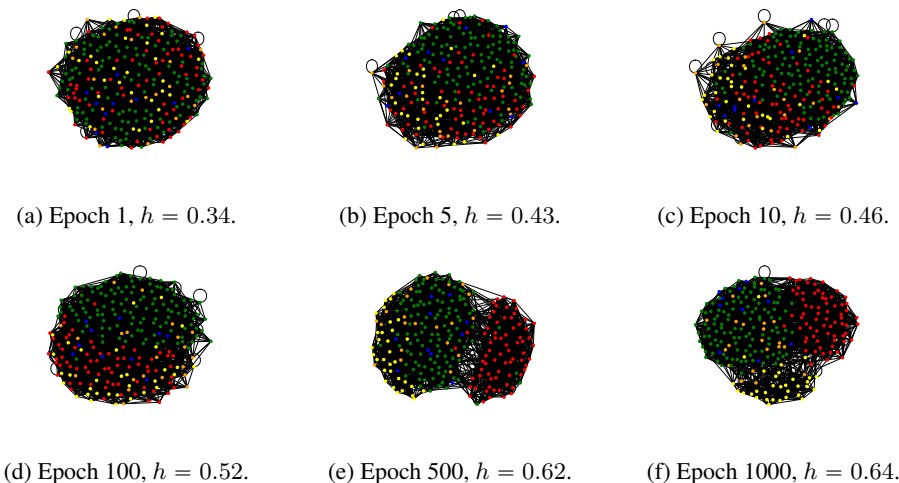

(a) Epoch 1, $h = 0.34$.     (b) Epoch 5, $h = 0.43$.     (c) Epoch 10, $h = 0.46$.

(d) Epoch 100, $h = 0.52$.     (e) Epoch 500, $h = 0.62$.     (f) Epoch 1000, $h = 0.64$.

Figure 9: Latent graph homophily level, $h$, evolution as a function of training epochs for Wisconsin. The latent graphs are produced during the training process for the GCN-dDGM*-EHS model with $k = 10$.

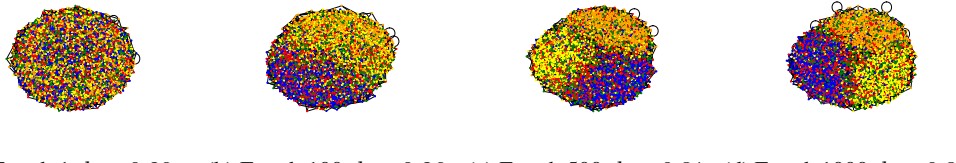

(a) Epoch 1, $h = 0.20$.    (b) Epoch 100, $h = 0.26$.    (c) Epoch 500, $h = 0.31$.    (d) Epoch 1000, $h = 0.32$.

Figure 10: Latent graph homophily level, $h$, evolution as a function of training epochs for Squirrel using the GCN-dDGM*-S model with $k = 3$. Recall that each node in the graph is represented with a point and each class is assigned a different color. In (a) there is no structure, after 1,000 epochs in (d) the algorithm has been able to organize the graph structure to separate some of the classes, but there is still a substantial amount of mixing.

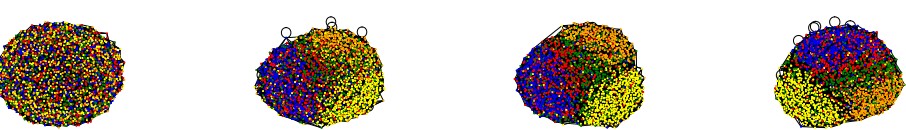

(a) Epoch 1, $h = 0.19$.    (b) Epoch 100, $h = 0.36$.    (c) Epoch 500, $h = 0.40$.    (d) Epoch 1000, $h = 0.42$.

Figure 11: Latent graph homophily level, $h$, evolution as a function of training epochs for Chameleon using the GCN-dDGM*-ES model with $k = 5$.

Lastly, we display the learned latent graphs using original heterophilic graph as inductive bias for both the Texas and Wisconsin datasets. In Figure 12 and Figure 13, we compare the final inferred latent graphs for the datasets when the original heterophilic dataset graph is used as inductive bias, against starting from a pointcloud. Models that use the dDGM module, which makes use of the original graph, are not able to achieve high homophily levels as compared to those using the dDGM* module and ignoring the original graph. This explains the difference in performance in Table 17 from Appendix E.4.

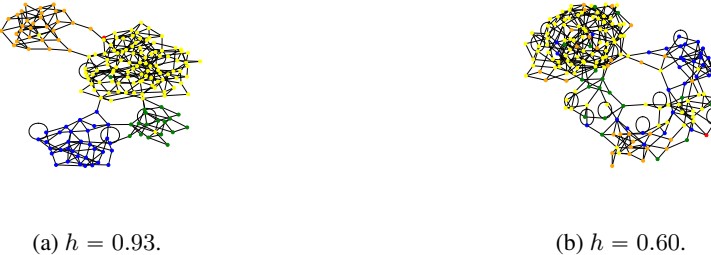

(a) $h = 0.93$.           (b) $h = 0.60$.

Figure 12: Comparison of final inferred latent graph (after 1,000 epochs of training) for the Texas dataset by (a) the GCN-dDGM*-E and (b) the GCN-dDGM-E model, both with $k = 2$.

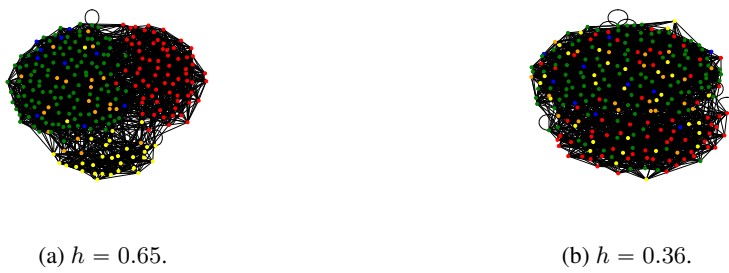

(a) $h = 0.65$.           (b) $h = 0.36$.

Figure 13: Comparison of final inferred latent graph (after 1,000 epochs of training) for the Wisconsin dataset by (a) the GCN-dDGM*-EHS and (b) the GCN-dDGM-EHS model, both with $k = 10$.

### F.2   LEARNED LATENT GRAPHS FOR REAL-WORLD DATASETS

Next we displayed the learned latent graph for the TadPole and the Aerothermodynamics datasets, see Figure 14, Figure 15, Figure 16, and Figure 17.

## G   MODEL ARCHITECTURES

For reproducibility, in this appendix we include a summary of all the models used to obtain the results discussed in this paper. This includes the GNN network archictectures as well as the internal structure of the dDGM modules.

### G.1   NETWORK ARCHITECTURES

In this section we provide the neural network architectures used for the experimental validation as well as other training specifications.

### G.1.1   NETWORKS FOR CLASSICAL GRAPH DATASETS

All neural network models used for homophilic graph datasets, MLP, GCN, and GCN-dDGMs, follow the architecture depicted in Table 21. We apply a learning rate of $lr = 10^{-2}$ and a weight decay of $wd = 10^{-4}$. Models are trained for about 1,500 epochs.

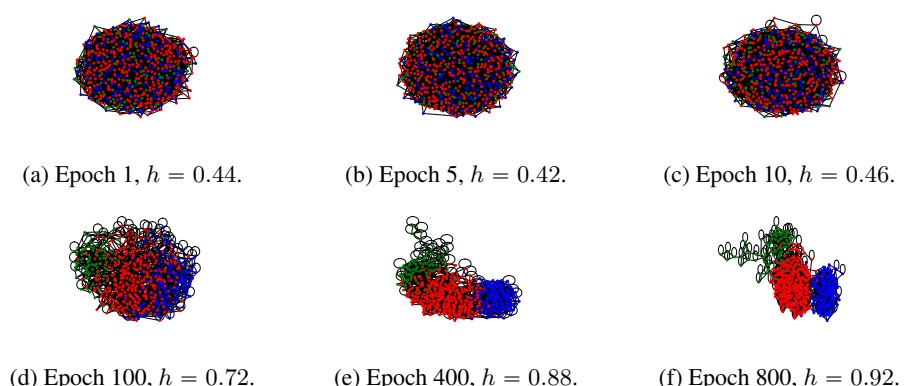

(a) Epoch 1, $h = 0.44$.      (b) Epoch 5, $h = 0.42$.      (c) Epoch 10, $h = 0.46$.

(d) Epoch 100, $h = 0.72$.      (e) Epoch 400, $h = 0.88$.      (f) Epoch 800, $h = 0.92$.

Figure 14: Latent graph homophily level, $h$, evolution as a function of training epochs for TadPole. The latent graphs shown here were obtain using the GCN-dDGM*-H model with $k = 3$.

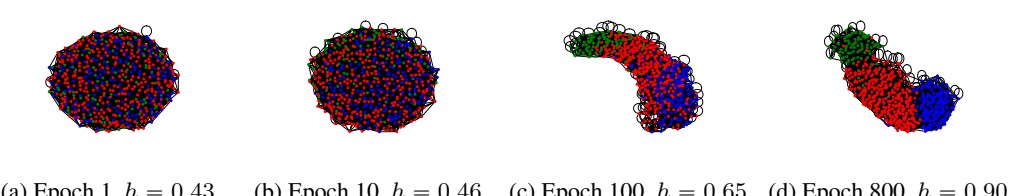

(a) Epoch 1, $h = 0.43$.    (b) Epoch 10, $h = 0.46$.    (c) Epoch 100, $h = 0.65$.    (d) Epoch 800, $h = 0.90$.

Figure 15: Latent graph homophily level, $h$, evolution as a function of training epochs for TadPole obtained using the GCN-dDGM*-EHS model with $k = 7$. Nodes with different colors correspond to different target classes. Starting from a unstructured graph in (a), the algorithm is able to generate a highly homophilic graph after 800 epochs in (d).

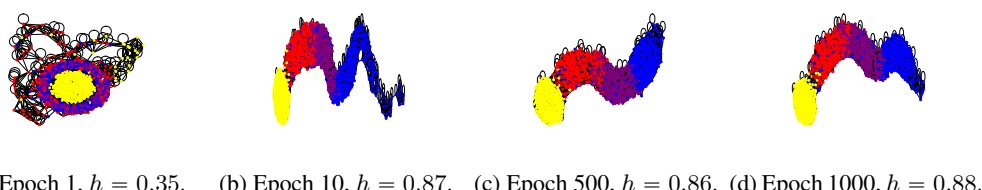

(a) Epoch 1, $h = 0.35$.    (b) Epoch 10, $h = 0.87$.    (c) Epoch 500, $h = 0.86$.    (d) Epoch 1000, $h = 0.88$.

Figure 16: Latent graph homophily level, $h$, evolution as a function of training epochs for Aerothermodynamics. The latent graphs shown here were obtain using the GAT-dDGM*-EHH model with $k = 7$.

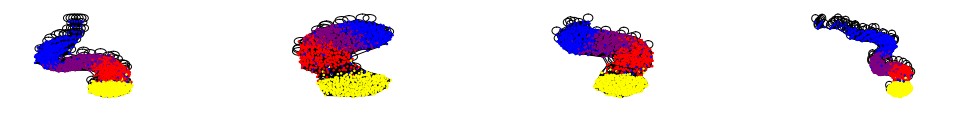

(a) Epoch 1, $h = 0.85$.    (b) Epoch 10, $h = 0.79$.    (c) Epoch 500, $h = 0.84$.    (d) Epoch 1000, $h = 0.87$.

Figure 17: Latent graph homophily level, $h$, evolution as a function of training epochs for Aerothermodynamics. The latent graphs shown here were obtain using the GAT-dDGM*-H model with $k = 7$.

Table 21: Summary of model architectures for experiments using classical graph datasets. Most experiments only use the first dDGM module.

| | | Model | | |
| --- | --- | --- | --- | --- |
| | | **MLP** | **GCN** | **GCN-dDGM** |
| No. Layer parameters | Activation | | Layer type | |
| | | N/A | N/A | dDGM |
| (No. features, 32) | ELU | Linear | Graph Conv | Graph Conv |
| | | N/A | N/A | dDGM |
| (32, 16) | ELU | Linear | Graph Conv | Graph Conv |
| | | N/A | N/A | dDGM |
| (16, 8) | ELU | Linear | Graph Conv | Graph Conv |
| (8, 8) | ELU | Linear | Linear | Linear |
| (8, 8) | ELU | Linear | Linear | Linear |
| (8, No. classes) | - | Linear | Linear | Linear |

### G.1.2 NETWORKS FOR HETEROPHILIC GRAPH DATASETS

The networks used for heterophilic datasets follow the architecture summarized in Table 22. We apply $lr = 10^{-2}$ and $wd = 10^{-3}$, and we train the models for about 1,000 epochs.

Table 22: Summary of model architectures for experiments using heterophilic graph datasets. The dDGM module uses different $k$ values.

| | | | Model | | |
| --- | --- | --- | --- | --- | --- |
| | | | **MLP** | **GCN** | **GCN-dDGM** |
| No. Layer parameters | BatchNorm | Activation | | Layer type | |
| | | | N/A | N/A | dDGM |
| (No. features, 16) | No | ELU | Linear | Graph Conv | Graph Conv |
| (16, 8) | No | ELU | Linear | Graph Conv | Graph Conv |
| (8, 8) | Yes | ELU | Linear | Linear | Linear |
| (8, No. classes) | No | - | Linear | Linear | Linear |

### G.1.3 NETWORKS FOR OGB-ARXIV

Table 23 shows the networks used for the OGB-Arxiv dataset. We use $lr = 10^{-3}$, $wd = 0$, and train for 100 epochs. For graph subsampling, we sample up to 1,000 neighbours per node and use a batch size of 1,000.

Table 23: Summary of model architectures for experiments for the OGB-Arxiv dataset using GATv2 diffusion layers. As specified in the table, $k = 20$ is used for all experiments.

| | | | Model | | |
| --- | --- | --- | --- | --- | --- |
| | | | **MLP** | **GATv2** | **GATv2-dDGM** |
| No. Layer parameters | BatchNorm | Activation | | Layer type | |
| | | | N/A | N/A | dDGM ($k = 20$) |
| (No. features = 128, 40) | No | SiLU | Linear | Graph Attention v2 | Graph Attention v2 |
| (40, 40) | No | SiLU | Linear | Graph Attention v2 | Graph Attention v2 |
| (40, 40) | No | SiLU | Linear | Graph Attention v2 | Graph Attention v2 |
| (40, 40) | Yes | SiLU | Linear | Linear | Linear |
| (40, 100) | Yes | SiLU | Linear | Linear | Linear |
| (100, No. classes = 40) | No | - | Linear | Linear | Linear |

### G.1.4 NETWORKS FOR OGB-PRODUCTS

Table 24 shows the networks used for the OGB-Products dataset. We use $lr = 10^{-2}$, $wd = 0$, and train for 30 epochs. For graph subsampling, we sample up to 200 neighbors per node and use a batch size of 1,000.

Table 24: Summary of model architectures for experiments for the OGB-Products dataset using GATv2 diffusion layers, with $k = 3$ for the dDGM module.

| | | | Model | | |
| | | | **MLP** | **GATv2** | **GATv2-dDGM** |
| No. Layer parameters | BatchNorm | Activation | | Layer type | |
| | | | N/A | N/A | dDGM ($k = 3$) |
| (No. features = 100, 100) | No | SiLU | Linear | Graph Attention v2 | Graph Attention v2 |
| (100, 100) | No | SiLU | Linear | Graph Attention v2 | Graph Attention v2 |
| (100, 47) | No | SiLU | Linear | Graph Attention v2 | Graph Attention v2 |
| (47, No. classes = 47) | No | - | Linear | Linear | Linear |

### G.1.5 NETWORKS FOR INDUCTIVE LEARNING

Table 25 summarizes the networks used for the QM9 and Alchemy datasets. We use $lr = 5 \times 10^{-2}$, $wd = 10^{-3}$, and train for 30 to 60 epochs.

Table 25: Summary of model architectures for experiments for the QM9 and Alchemy datasets. These models incorporate a global mean pool layer since predictions are at the graph level, rather than at the node level as in all other datasets.

| | | | Model | | |
| | | | **MLP** | **GCN** | **GCN-dDGM** |
| No. Layer parameters | BatchNorm | Activation | | Layer type | |
| | | | N/A | N/A | dDGM |
| (No. features, 20) | Yes (GraphNorm) | SiLU | Linear | Graph Conv | Graph Conv |
| (20, 20) | Yes (GraphNorm) | SiLU | Linear | Graph Conv | Graph Conv |
| | | | | Global Mean Pool | |
| (20, 20) | No | SiLU | Linear | Linear | Linear |
| (20, 20) | No | SiLU | Linear | Linear | Linear |
| (20, No. prediction targets) | No | - | Linear | Linear | Linear |

### G.1.6 NETWORKS FOR BRAIN IMAGING

Table 26 summarizes the networks used for the TadPole dataset. We use $lr = 10^{-3}$ and $wd = 2 \times 10^{-4}$, and train for about 800 epochs.

Table 26: Summary of model architectures for TadPole.

| | | | Model | | |
| | | | **MLP** | **GCN-dDGM** | **GAT-dDGM** |
| No. Layer parameters | BatchNorm | Activation | | Layer type | |
| | | | N/A | dDGM | dDGM |
| (No. features, 32) | No | ELU | Linear | Graph Conv | Graph Attention |
| (32, 16) | No | ELU | Linear | Graph Conv | Graph Attention |
| (16, 8) | No | ELU | Linear | Graph Conv | Graph Attention |
| (8, 8) | Yes | ELU | Linear | Linear | Linear |
| (8, 8) | Yes | ELU | Linear | Linear | Linear |
| (8, No. classes) | No | - | Linear | Linear | Linear |

### G.1.7 NETWORKS FOR AEROSPACE ENGINEERING

Table 27 summarizes the networks used for the Aerothermodynamics dataset. We use $lr = 10^{-2}$ and $wd = 0$, and train for about 1,000 epochs.

Table 27: Summary of model architectures for Aerothermodynamics.

| | | | Model | | |
| --- | --- | --- | --- | --- | --- |
| | | | **MLP** | **GCN-dDGM** | **GAT-dDGM** |
| No. Layer parameters | BatchNorm | Activation | Layer type | | |
| | | N/A | N/A | dDGM | dDGM |
| (No. features, 32) | No | ELU | Linear | Graph Conv | Graph Attention |
| (32, 16) | No | ELU | Linear | Graph Conv | Graph Attention |
| (16, 8) | No | ELU | Linear | Graph Conv | Graph Attention |
| (8, 8) | Yes | ELU | Linear | Linear | Linear |
| (8, 8) | Yes | ELU | Linear | Linear | Linear |
| (8, No. classes) | No | - | Linear | Linear | Linear |

## G.2 DDGM ARCHITECTURES

The dDGM and dDGM* internal parameterized mapping functions are slightly modified for different datasets. All architectures are recorded in Table 28, 29, 30, 31, 32, 33, and 34.

Table 28: dDGM* and dDGM architectures for classical homophilic datasets.

| | | **dDGM*** | **dDGM** |
| --- | --- | --- | --- |
| No. Layer parameters | Activation | Layer type | |
| (No. features, 32) | ELU | Linear | Linear |
| (32, 16 per model space) | ELU | Linear | Graph Conv |
| (16 per model space, 4 per model space) | Sigmoid | Linear | Graph Conv |

Table 29: dDGM* and dDGM architectures for heterophilic datasets.

| | | | **dDGM*** | **dDGM** |
| --- | --- | --- | --- | --- |
| No. Layer parameters | BatchNorm | Activation | Layer type | |
| (No. features, 32) | Yes | ELU | Linear | Linear |
| (32, 4 per model space) | Yes (dDGM*)/ No (dDGM) | ELU | Linear | Graph Conv |
| (4 per model space, 4 per model space) | Yes (dDGM*)/ No (dDGM) | Sigmoid | Linear | Graph Conv |

Table 30: dDGM* and dDGM architectures for OGB-Arxiv.

| | | | **dDGM*** | **dDGM** |
| --- | --- | --- | --- | --- |
| No. Layer parameters | BatchNorm | Activation | Layer type | |
| (No. features, 40) | Yes | SiLU | Linear | Linear |
| (40, 32 per model space) | Yes (dDGM*)/ No (dDGM) | SiLU | Linear | Graph Conv |
| (32 per model space, 16 per model space) | Yes (dDGM*)/ No (dDGM) | SiLU | Linear | Graph Conv |

Table 31: dDGM* and dDGM architectures for OGB-Products.

| | | | **dDGM*** | **dDGM** |
| --- | --- | --- | --- | --- |
| No. Layer parameters | BatchNorm | Activation | Layer type | |
| (No. features, 32 per model space) | Yes (dDGM*)/ No (dDGM) | SiLU | Linear | Graph Conv |
| (32 per model space, 16 per model space) | Yes (dDGM*)/ No (dDGM) | SiLU | Linear | Graph Conv |

Table 32: dDGM* and dDGM architectures for QM9 and Alchemy.

| | | | **dDGM*** | **dDGM** |
| --- | --- | --- | --- | --- |
| No. Layer parameters | BatchNorm | Activation | Layer type | |
| (No. features, 3 per model space) | Yes | SiLU | Linear | Graph Conv |
| (3 per model space, 3 per model space) | Yes | SiLU | Linear | Graph Conv |

Table 33: dDGM* and dDGM architectures for TadPole.

|  |  |  | dDGM* | dDGM |
| --- | --- | --- | --- | --- |
| No. Layer parameters | BatchNorm | Activation | Layer type | |
| (No. features, 16 per model space) | Yes (dDGM*)/ No (dDGM) | ELU | Linear | Graph Conv |
| (16 per model space, 4 per model space) | Yes (dDGM*)/ No (dDGM) | Sigmoid (dDGM*)/ ELU (dDGM) | Linear | Graph Conv |

Table 34: dDGM* and dDGM architectures for Aerothermodynamics.

|  |  |  | dDGM* | dDGM |
| --- | --- | --- | --- | --- |
| No. Layer parameters | BatchNorm | Activation | Layer type | |
| (No. features, 32) | No | ELU | Linear | Linear |
| (32, 16 per model space) | Yes (dDGM*)/ No (dDGM) | ELU | Linear | Graph Conv |
| (16 per model space, 4 per model space) | Yes (dDGM*)/ No (dDGM) | Sigmoid (dDGM*)/ ELU (dDGM) | Linear | Graph Conv |

