# OpenReview forum: "Latent Graph Inference using Product Manifolds"
_ICLR.cc/2023/Conference — ICLR 2023 poster_

### Official Review · Reviewer_R63Y · 2022-10-17

**Confidence:** 4
**Correctness:** 3
**Technical Novelty And Significance:** 2
**Empirical Novelty And Significance:** 1
**Recommendation:** 5

**Clarity, Quality, Novelty And Reproducibility:**

As is discussed in S2, the paper has good clarity and quality. In specific, the research problem and proposed method is well formulated. The paper is well written.

**Strength And Weaknesses:**

### Strengths:

S1: A great number of experiments are conducted.
The authors conduct the experiments on various of datasets, including homophilic, heterophilic, and real-world application-based datasets. Visualization experiments are also used to exhibit the quality of the learned graph structure.

S2: The paper is well presented and written.
The definitions and notations in the paper are explicit. The proposed method is clearly described. The paper is written in good quality.

### Weaknesses:

W1: The novelty of this paper is limited.
As is described by the authors, the proposed method is based on dDGM (Kazi et al. 2022). The main difference between dDGM and the proposed method is that, dDGM mainly learns on Euclidean, while the proposed method learns on both Euclidean and other non-constant curvature spaces. However, in Section 5.1.2 and Table 3 of paper (Kazi et al. 2022), this paper already discusses the potential of applying dDGM to non-Euclidean geometry latent space and conducts an experiment on Hyperbolic space. Such a previous discussion makes the contribution of this paper less significant. In this case, the authors are suggested to better clarify the novelty and contribution of this paper.

W2: Some critical related works lack in the literature review.
This paper focuses on learning latent graph structure from raw data, which is described as “the latent graph inference problem” in this paper. However, in the community of GNNs, there is a branch of works aims to address “graph structure learning problem” which has a consistent learning target with this paper. In this case, it’s of great significance to review the related papers about graph structure learning and discuss the similarity/difference between the proposed method and these works. For this topic, the authors are suggested to refer recent survey [1] and some classic methods (e.g., LDS-GNN [2], IDGL [3], and ProGNN [4]).

[1] Zhu, Yanqiao, et al. "Deep graph structure learning for robust representations: A survey." arXiv preprint arXiv:2103.03036 (2021).

[2] Franceschi, Luca, et al. "Learning discrete structures for graph neural networks." International conference on machine learning. PMLR, 2019.

[3] Chen, Yu, Lingfei Wu, and Mohammed Zaki. "Iterative deep graph learning for graph neural networks: Better and robust node embeddings." Advances in neural information processing systems 33 (2020): 19314-19326.

[4] Jin, Wei, et al. "Graph structure learning for robust graph neural networks." Proceedings of the 26th ACM SIGKDD international conference on knowledge discovery & data mining. 2020.

W3: The experiments seriously lack baselines for comparison.
In the experiments, the proposed method is only compared to MLP and GCN, making the results not convincing enough. Readers may be curious about how the proposed method performs compared to state-of-the-art methods. At least, the authors should add the original dDGM for comparison (is that equal to GCN-dDGM-E?). Moreover, some representative reviewed methods (e.g., DGCNN and DGMs) and the graph structure learning methods (listed in W2) should also be considered.


**Summary Of The Paper:**

This paper aims to address the latent graph inference problem, i.e., inferring the intrinsic graph structure from point cloud-like data where connection is not available in the original data. Based on the discrete Differentiable Graph Module (dDGM) proposed by previous work, this paper extends dDGM with Riemannian geometry by calculating distance (which is used for edge modelling) in not only Euclidean space but also hyperboloid/hypersphere spaces. Extensive experiments are conducted to show the effectiveness of the proposed method.

**Summary Of The Review:**

Clarity/Quality:

As is discussed in S2, the paper has good clarity and quality. In specific, the research problem and proposed method is well formulated. The paper is well written.

Novelty:

As is discussed in W1, the novelty of this paper is limited. In specific, compared to the basic model dDGM, the technical improvement of the proposed method is relevantly minor.

Reproducibility:

This paper has great reproducibility. The experimental details are described. The source code is not provided.

---

> ### Author Response · Authors · 2022-11-09
> **Response to W1**
>
> We would like to thank the reviewer for the thorough review. We are glad to hear that the reviewer appreciated the extensive experimental results and enjoyed the visualizations. Below we address the concerns in order.
>
> W1 answer) We appreciate the concern related to the novelty of the work. We would like to clarify our contributions. The original paper does mention the Poincare Ball as a possible embedding space (this is a stereographic projection of the hyperboloid). However, they do not use a principled way to transfer the Euclidean output feature vectors of the GNN layers to the non-Euclidean space. We introduce exponential maps tangent to the poles of the hyperboloid and the hypersphere to project the points to the non-Euclidean spaces. Once the points have been projected, we can calculate meaningful distances between the points residing on hyperbolic space or spherical space. Without using the exponential map, one is naively applying the Poincare ball distance on an embedding space which is not actually hyperbolic. In our work we consider non-Euclidean spaces with both positive and negative curvature, using the model spaces: the hypersphere and the hyperboloid. The original work only worked with the Euclidean plane which has 0 curvature (and with the Poincare ball, but as mentioned this was not done correctly: indeed in the original paper they need to clip the norm of the latent vectors when using the Poincare ball, this is likely due to not using the exponential map). Hence, we cover all possible values of the curvature and introduce a principled way to apply latent graph inference in non-Euclidean spaces. This can be considered as a key contribution.
>
> Moreover, we introduce a conceptual leap as compared to the original formulation in Kazi et. al. The authors in Kazi et. al understand the embedding space as a latent space from which it is possible to generate edge probabilities. However, in this work, inspired by the manifold hypothesis, we aim at generating more complex manifolds beyond model spaces of constant curvature.  We do this from the point of view that a graph is a discretized version of a manifold, rather than thinking of probabilities. Many machine learning algorithms rely on the manifold hypothesis which suggests that although the data in the observed space may reside in a Euclidean hypercube of high-dimensionality, the samples can actually be explained by an embedded low-dimensional submanifold within the original space. In general, such manifold can have a complicated structure, varying curvature and even disjoint density regions. Using model spaces for which the curvature is constant (even if it is non-Euclidean) is likely to distort the latent representations. In this work, we introduce a computationally tractable approach to try to approximate the underlying manifold and use it to generate a latent graph, which can be understood as a discretization of the continuous latent manifold. The main problem is that generating a complex manifold is computationally difficult. There is no straightforward way to obtain exponential maps and geodesics for an arbitrary manifold, even less so to make the whole process differentiable. Using product manifolds it is possible to leverage the closed form solutions of independent model spaces for exponential maps and distances, so that we can generate complex manifolds for which we can obtain closed form solutions which are differentiable and hence can be optimized along the rest of the network. This is discussed in Section 3.2 and 3.3 of the main paper, and further details on the topic are provided in Appendix B and Appendix C.2. This can be considered as a further contribution. Note also that the original paper by Kazi et al, optimizes the dDGM network parameters that transform the inputs to the module (they tune the network parameters), but importantly they do not optimize the curvature of the manifold of the latent space. In our work we try to model in a fully-differentiable fashion the underlying embedded manifold: we deform the embedded space to better represent the data by updating the curvature of the space based on the downstream task performance.
>
> Lastly, we also demonstrate the applicability of this approach to heterophilic datasets (which the original paper did not consider at all), and present an interesting finding: the latent graph inference  system allows us to generate latent homophilic graphs if we optimize it in conjunction with layers such as GCN or GAT, which assume the underlying graph is homophilic, that is, they use a homophilic inductive bias.
>
> We continue answering the rest of the concerns in the next comment due to the character limit.

---

> > ### Author Response · Authors · 2022-11-09
> > **Response to W2 and W3**
> >
> > W2 answer) Thank you for pointing out the papers. We have added the recommended references in the Related Work section (Section 2.1).
> >
> > W3 answer)  Thank you for highlighting the concerns regarding baselines. In the text it may not be so clear that the Euclidean dDGM module is equivalent to the original dDGM by Kazi et al. We have amended the submission to clarify this point. Indeed GCN-dDGM-E is equivalent to the dDGM module by Kazi et al, since it uses the Euclidean plane as embedding space. This allows us to compare our new approach against the original paper.
> >
> > We have added the following clarifications in the main text.
> >
> > “The model spaces are denoted as: Euclidean (dDGM-E/dDGM*-E, which is equivalent to the original architecture used by Kazi et al, hyperbolic (dDGM-H/dDGM*-H), and spherical space (dDGM-S/dDGM*-S).” in Section 4.
> >
> > “Note that the models which use the Euclidean plane as embedding space (denoted with an E in the table) are equivalent to those presented in Kazi et al” in the caption of Table 2.
> >
> > Furthermore, in the original submission we included the MLP and GCN baselines. The MLP considers nodes independently, that is, it doesn’t use any graph information for prediction. The GCN uses the original graph, but not a latent graph. Lastly, our method uses a latent graph for improved performance. If the dataset has an original graph, this is used as an inductive bias. If not, we generate a latent graph by only using the information from the pointcloud features. In the case of pointcloud data, the GCN baseline is not applicable because there is no input graph. In general, we expect using an optimized latent graph to give improved results as compared to these baselines.
> >
> > Question on Reproducibility: The source code is not provided.  Answer) As mentioned in the text (Section 1), to make this work accessible to the wider machine learning community, we have created a new PyTorch Geometric layer.
> >
> > Thank you for your kind review. Please let us know if we can do anything else to address your concerns.

---

> > > ### Comment · Reviewer_R63Y · 2022-11-17
> > > **Feedback**
> > >
> > > I appreciate the reply from the authors. The reply partly addresses my concerns. In concrete, the novelty of the proposed method (compared to Kazi et al.) has been clearly highlighted (W1). The literature review has been better refined (W2).
> > >
> > > However, I am still concerned about the lack of baselines in the experiments. As graph structure learning methods have developed well in these years and are highly related to the learning scenario in this paper, it is of great significance to compare the graph structure learning methods with the proposed method. I know the authors have compared the performance of different variants of the proposed method, but they are not enough to show the superior of the proposed method over existing state-of-the-art methods.
> > >
> > > I'm happy to give a higher recommendation score if the authors address the aforementioned concern.

---

> > > > ### Author Response · Authors · 2022-11-17
> > > > **Baselines Added**
> > > >
> > > > Dear Reviewer,
> > > >
> > > > Thank you for your response. We are happy to hear that we have addressed some of your concerns. We have now added the following baselines presented in the literature in the newly uploaded document: LDS-GNN, IDGL, IDGL-ANCH, Pro-GNN, Pro-GNN-fs, and GCN-Jaccard.
> > > >
> > > > Please let us know if we can do anything else to address your concerns. Thank you for your help.

---

> > > > > ### Author Response · Authors · 2022-11-17
> > > > > **Baselines in the main text (location)**
> > > > >
> > > > > The mentioned baselines can be found in Table 2.

---

> > > > > ### Comment · Reviewer_R63Y · 2022-11-18
> > > > > **Baselines**
> > > > >
> > > > > Dear authors,
> > > > >
> > > > > I'm happy to hear that you've updated the baselines. However, I found some issues with the updated results:
> > > > >
> > > > > 1) It seems that the results are copied from the original papers of IDGL and Pro-GNN. However, these two papers use different train/val/test splits, indicating that these results cannot be compared directly. I suggest the authors reproduce these methods following the train/val/test split of your setting, ensuring that the comparison is reasonable and reliable.
> > > > >
> > > > > 2) The results on most of the datasets are missing, making the comparison not sufficient enough.
> > > > >
> > > > > To sum up, I don't think the current comparison is sound and reasonable to show the effectiveness of the proposed method.

---

> > > > > > ### Author Response · Authors · 2022-11-18
> > > > > > **Baselines**
> > > > > >
> > > > > > Dear Reviewer,
> > > > > >
> > > > > > Thank you for your response. We understand your concerns and endeavour to address them in a future revised version. Given time constraints it is really challenging to try to reproduce the results for all datasets and all benchmarks (note that we have also included heterophilic datasets which the previous works did not discuss). Moreover, the implementations of different baselines that can be found online use different frameworks (PyTorch,Tensorflow,etc).
> > > > > >
> > > > > > For example, the paper that you suggested ‘Iterative Deep Graph Learning for Graph Neural Networks: Better and Robust Node Embeddings’ does not include results for all datasets and all models either, we cite from caption of Table 1: ‘The dash symbol indicates that reported results were unavailable or we were not able to run the experiments due to memory issue.’ In our case, we addressed the scalability issue using symbolic matrices instead of dense PyTorch matrices. Being able to scale the original methods to larger datasets may hence require additional modifications to the source code.
> > > > > >
> > > > > > However, we have compared our method to that in Kazi et. al (2022) and to vanilla benchmarks such as GCNs and MLPs. We hope that the literature benchmarks we have included so far are indicative of the effectiveness of our method. We do appreciate your concern and we commit to comparing the other methods you suggested and running experiments for all the datasets before the camera ready submission.
> > > > > >
> > > > > > Additionally, we would like to highlight that the main contribution of the paper is showing that previous latent graph inference/graph structure learning methods can benefit from incorporating Riemannian Geometry to their underlying latent structure. In particular, in this work we demonstrate that using product manifolds is beneficial as compared to the original dDGM module which only operates in Euclidean space. This concept could potentially be generalized to other graph structure learning approaches.
> > > > > >
> > > > > > Thank you for your help and meaningful insights.

---

> > > > > > > ### Comment · Reviewer_R63Y · 2022-11-19
> > > > > > > **Baselines**
> > > > > > >
> > > > > > >
> > > > > > > Thanks for your comments. I think scientific research requires convincing validation of the experimental results with a fair and thorough comparison. The comparison of the proposed method with IDGL and Pro-GNN seems necessary to show the performance of the proposed algorithm. However, the comparison should be conducted in a fair setting (e.g., with the same training/validation/test split). Copying the results from other papers with different settings is meaningless and is even misleading to future research in the community.
> > > > > > >
> > > > > > > I understand the promise of adding the results to the paper in a camera-ready version. However, it is unsure if the proposed method will still outperform baselines in the same settings. So it seems to me that the paper is not ready at this stage for publishing at ICLR.

---

> ### Comment · Reviewer_R63Y · 2022-12-07
> **Update**
>
> I have updated my ratings after reconsidering the authors' responses and other reviewers' comments. I do see the value of the paper and I think more baselines will enhance the paper.

---

### Official Review · Reviewer_vGZV · 2022-10-25

**Confidence:** 3
**Correctness:** 4
**Technical Novelty And Significance:** 4
**Empirical Novelty And Significance:** 3
**Recommendation:** 8

**Clarity, Quality, Novelty And Reproducibility:**

This paper is clearly written, and the authors created a new PyTorch layer to help with reproducibility.

**Strength And Weaknesses:**

Strength: Modeling data points on different manifolds to embed more complicated relationship information into the graph is an interesting idea, and seems to work well for experiments with heterophilic datasets. Visualization of latent graph evolution in Figure 2 is intriguing.

Weakness: Notations used in section 3.1. can be hard to understand. In section 4.2., if the data doesn't come with a graph and are just pointclouds, why would one decide to use GNNs in the first place?

**Summary Of The Paper:**

This paper generalizes discrete Differentiable Graph Module (dDGM) to learn latent graphs in the product space of Euclidean planes, hyperboloids and hyperspheres of constant curvature. The learned graphs are then used for graph neural networks. Experiments are done on heterophilic and homophilic datasets, as well as some real-world data without a graph given as a prior.

**Summary Of The Review:**

It's an interesting approach to latent graph learning that builds on prior work on differentiable Riemannian manifolds. The experimental results are promising.

---

> ### Author Response · Authors · 2022-11-09
> **Response to Reviewer regarding Pointclouds**
>
> Dear Reviewer,
>
> We would like to thank you for your review. We are glad to hear that you have found our work to be interesting and intriguing.
>
> We now address your point regarding the applicability to pointclouds. Although some datasets are presented to us in the form of pointclouds and without a directly accessible graph structure, in this work we leverage the hypothesis that there exists an underlying but unknown latent graph that can improve downstream performance. When you work only with pointclouds you evaluate each point in isolation, whereas using a latent graph allows the model to share information between data points and exploit valuable relationships between them.
>
> Indeed, the Tadpole and Aerothermodynamics datasets do not have an input graph, but we can see that using the latent graph inference system improves predictive performance as compared to using a MLP, which only considers the feature vectors of each node in isolation. In general, in a pointcloud some points could be more relevant to each other. Hence, learning latent connectivity information is beneficial and useful for solving the task at hand. We consider this as a good feature of our model, since it can handle data without any input graph information (such as pointclouds), and learn better representations compared to MLPs, for example.
>
> Furthermore, imposing an embedding space of constant curvature can deform the underlying structure of the data, which in general, can have complex behaviour with both regions of positive, negative, and zero curvature. Using product manifolds we can learn a richer latent manifold that can better model the dynamics of the system, plus also optimise curvature, in a fully-differentiable fashion. Moreover, our approach allows us to model graphs based on latent manifolds, rather than inferring a manifold or analysing the curvature of an already existing graph. One of the main conceptual leaps in this work is that we understand the latent/embedding space as being a complicated manifold from which we can obtain a latent graph (a discretized form of the continuous manifold), rather than seen the embedding space as just a latent representation from which we generate probabilities.
>
> Thank you for your kind review.

---

### Official Review · Reviewer_DMwU · 2022-10-27

**Confidence:** 2
**Correctness:** 3
**Technical Novelty And Significance:** 3
**Empirical Novelty And Significance:** 3
**Recommendation:** 6

**Clarity, Quality, Novelty And Reproducibility:**

The paper is well-organized. The novelty is sufficient and the contributions are significant.

**Strength And Weaknesses:**

Strength:

1. The authors provide a novel method to generalize the discrete differentiable graph module for learning latent graphs.

2. It is interesting to incorporate Riemannian geometry into the model to generate more complex embedding spaces. The resultant performance is improved.

Weakness:

1. It is unknown the scalability of the proposed method.



**Summary Of The Paper:**

In this paper, the authors proposed to generalize the discrete Differentiable Graph Module (dDGM) for latent graph learning, by incorporating the Riemannian geometry into the model and generating more complex embedding spaces. The authors further propose a computationally tractable approach to produce product manifolds of constant curvature model spaces that can encode latent features of varying structures. The numerical results demonstrate that the proposed method outperforms the original dDGM model.




**Summary Of The Review:**

The authors present a new method to generalize the discrete differentiable graph module for learning latent graphs, which could be interesting to people working on graph neural networks. Such a method could provide a way to learn a better graph topology, which helps to improve the performance of graph neural networks in the downstream task. One concern to be solved is the scalability of the proposed method.

---

> ### Author Response · Authors · 2022-11-09
> **Response to Reviewer regarding Scalability Concerns**
>
> Dear Reviewer,
>
> We would like to thank you for your careful review.
>
> We agree with the reviewer on the importance of analyzing the scalability of the method. ‘Section 5: Discussion and Conclusion’ discusses suggestions regarding this topic. Using symbolic matrices partially addresses the scalability problem. We have included an extensive analysis regarding the runtime speedup that can be obtained using symbolic matrices in Appendix C as compared to using fully populated (dense) standard PyTorch matrices.
>
> This was included in Appendix C and not in the main text since scalability concerns are inherent to the dDGM module, not specific to our technique, and not the major focus of our paper.  However, we considered this to be an important problem, but due to the page limit, we could only add it in Appendix C. More specifically, results for these experiments can be found in Table 5, 6, 7 and 8.
>
> The main contribution of this paper is to use Cartesian products of constant curvature model spaces to construct the latent graph which captures more complex latent relationships between the nodes. Scalability is mainly affected by the number of nodes in the input graph, regardless of whether Riemannian geometry is applied or not to the embedding space. In Section 5, under future work, we discuss this topic and propose plausible future research directions to try to alleviate this problem by proposing modifications to the dDGM module.
>
> Thank you for your kind review.

---

> > ### Comment · Reviewer_DMwU · 2022-11-17
> > **Thank you for the response!**
> >
> > Thank you for the detailed response. I would like to recommend that the authors could add more scalability-related discussions or experiments in the main manuscript.

---

> > > ### Author Response · Authors · 2022-11-17
> > > **Added more scalability-related discussions**
> > >
> > > Dear Reviewer,
> > >
> > > Thank you for your response. Due to the page limit we are unable to fit additional results from the appendix in the main text. However, we have followed your advice and included an extended discussion on scalability. In particular, the following sections may be of interest:
> > >
> > > In Section 4 ‘Experiments and Results’, the last paragraph before Section 4.1 starts reads:
> > >
> > > “Note that we only use a single latent graph inference module per neural network, that is, networks diffuse information based on only one latent graph. This is in line with previous work (Kazi et al. 2022). Additionally, in Appendix E.1, we investigate the effect of leveraging multiple latent graphs in the same network and conclude that in general it is better to use a single latent graph due to computational efficiency and diminishing returns. The study regarding computational efficiency can be found in Appendix C.3. In particular, we compare the runtime speedup obtained using symbolic matrices as compared to standard dense PyTorch matrices. We observe that as more product manifolds and dDGM modules are included, the runtime speedup obtained using symbolic matrices becomes increasingly large. Moreover, without using symbolic matrices standard GPUs (we use NVIDIA P100 and Tesla T4) run out of memory for datasets with $\mathcal{O}(10^{4})$ nodes such as PubMed, Physics, and CS. Hence, we recommend using symbolic matrices to help with scalability. Model architecture descriptions for all experiments can be found in Appendix G.”
> > >
> > > In Section 5 ‘Discussion and Conclusion’ the following suggestions to improve scalability are also included:
> > >
> > > Lastly, there are a few limitations intrinsic to the dDGM module, irrespective of the product manifold embedding approach introduced in this work. Firstly, although utilizing symbolic matrices can help computational efficiency (Appendix C.3), the method still has quadratic complexity. Kazi et. al (2022) proposed computing probabilities in a neighborhood of the node and using tree-based algorithms to reduce it to $\mathcal{O}(n\log n)$ […] Another avenue to help with scalability, improve computational complexity, and facilitate working with large-scale graphs would be to use a hierarchical perspective. Inspired by brain interneurons, we could introduce fictitious connector inducing nodes in different regions of the graph, use those nodes to summarize different regions of large graphs, and apply the kNN algorithm or the Gumbel Top-k trick to the fictitious connector inducing nodes. This way the computational complexity would still be quadratic, but proportional to the number of interconnectors. Similar techniques have been applied to Gaussian Processes and Set Transformers.
> > >
> > > (some citations in the original text have been omitted in this response)
> > >
> > > Please let us know if we can do anything else to address your concerns. Thank you for your help.

---

### Public Comment · ~Benedek_Andras_Rozemberczki1 · 2022-11-05
**Misattribution of datasets**

The paper misattributed the authorship of the Chameleons and Squirrels datasets. These datasets were proposed in this ICLR submission:

https://openreview.net/forum?id=HJxiMAVtPH

The Pei et al. paper cited by the authors took the Squirrel and Chameleons datasets and used those for benchmarking, but had nothing to do with the creation of the datasets. The correct citation for the paper which proposed the datasets is:

```bibtex
>@article{musae,
          author = {Rozemberczki, Benedek and Allen, Carl and Sarkar, Rik},
          title = {{Multi-Scale Attributed Node Embedding}},
          journal = {Journal of Complex Networks},
          volume = {9},
          number = {2},
          year = {2021},
}
```

---

> ### Author Response · Authors · 2022-11-08
> **Citation Added to Revised Version**
>
> Dear Benedek Andras Rozemberczki,
>
> We are sorry for this misunderstanding and thank you for letting us know, we have added the citation to the revised document.

---

### Decision · Program_Chairs · 2023-01-20

**Decision:**

Accept: poster

**Justification For Why Not Higher Score:**

The novelty is only incremental: this is an extension of dDGM to Riemanian geometry.

**Justification For Why Not Lower Score:**

The authors provide an interesting framework for embedding data using a product of manifolds.

**Metareview: Summary, Strengths And Weaknesses:**

__Summary.__ This paper considers the problem of latent graph learning. More specifically, they assume that they do not have access to the adjacency matrix and want to study how to learn the latent graph. This has applications for problems in which the observed adjacency matrix might be noise for instance, and to alleviate some of the issues that methods like GCNs or GAT have encountered when deployed on heterophilic datasets. To this end, they propose to enrich the discrete Differentiable Graph Module (dDGM) proposed by Kazi et al. (2022) (which is essentially, a graph construction method that relies of a stochastic relaxation of the kNN rule) with Riemannian geometry. Latent data is embedded using product manifolds of model spaces. The authors check the validity of their method through an extensive set of experiments.

__Summary of the reviews.__ The topic has been deemed interesting by the reviewers, and the contribution significant enough to warrant acceptance at ICLR. Initial reservations concerning the paper concerned (a) its lack of scalability (an issue that, after the rebuttal, the authors have discussed at length in the conclusion, and which is also shared with the dDGM method that they build upon) and (b) its lack of novelty, since the use of hyperbolic manifolds had been alluded to in the original dDGM paper. For the latter part, the authors highlight issues with the way hyperbolic manifolds were originally used, and highlight the fact that their more principled framework allows them to work with a variety of manifold. Overall, the authors seem to make a good case for their method: it does seem to make an improvement on DGM.

Notes: current issues with the approach: (a) The tables are unreadable. The font is too small, and the abbrevations are almost unintelligible. (b) The results are not impressive on heterophilic data. The question has to be raised: with this amount of variability, it is unclear whether the difference in proposed architecture is actually significant (especially after multiple hypothesis correction).

For these reasons, we recommend a (very weak) accept of this paper for a poster presentation.

**Note From Pc:**

if the above contains the word "oral" or "spotlight" please see: "oral" presentation means -> notable-top-5% and "spotlight" means -> notable-top-25%. As stated in our emails, we are disassociating presentation type from AC recommendations

**Summary Of Ac-Reviewer Meeting:**

While originally a borderline paper, the reviewers raised their scores after the rebuttal, thus rendering a meeting unnecessary.